# Regulation of the linear ubiquitination of STAT1 controls antiviral interferon signaling

Yibo Zuo[1,2], Qian Feng[1,2], Lincong Jin[1,2], Fan Huang[1,2], Ying Miao[1,2], Jin Liu[1,2], Ying Xu[3], Xiangjie Chen[1,2], Hongguang Zhang[1,2], Tingting Guo[1,2], Yukang Yuan[1,2], Liting Zhang[1,2], Jun Wang[3] & Hui Zheng [1,2✉]

Linear ubiquitination is a critical regulator of inflammatory signaling pathways. However, linearly ubiquitinated substrates and the biological significance of linear ubiquitination is incompletely understood. Here, we show that STAT1 has linear ubiquitination at Lys511 and Lys652 residues in intact cells, which inhibits STAT1 binding to the type-I interferon receptor IFNAR2, thereby restricting STAT1 activation and resulting in type-I interferon signaling homeostasis. Linear ubiquitination of STAT1 is removed rapidly by OTULIN upon type-I interferon stimulation, which facilitates activation of interferon-STAT1 signaling. Furthermore, viruses induce HOIP expression through the NF-κB pathway, which in turn increases linear ubiquitination of STAT1 and thereby inhibits interferon antiviral response. Consequently, HOIL-1L heterozygous mice have active STAT1 signaling and enhanced responses to type-I interferons. These findings demonstrate a linear ubiquitination-mediated switch between homeostasis and activation of type-I interferon signaling, and suggest potential strategies for clinical antiviral therapy.

[1] International Institute of Infection and Immunity, Institutes of Biology and Medical Sciences, Soochow University, 215123 Suzhou, Jiangsu, China. [2] Jiangsu Key Laboratory of Infection and Immunity, Soochow University, 215123 Suzhou, Jiangsu, China. [3] Department of Intensive Care Medicine, the First Affiliated Hospital of Soochow University, Soochow University, 215123 Suzhou, Jiangsu, China. ✉email: huizheng@suda.edu.cn

Protein ubiquitination has emerged as a key mechanism for controlling cellular signaling pathways[1–4]. The ubiquitin molecule harbors seven internal lysine residues (Lys6, Lys11, Lys27, Lys29, Lys33, Lys48, or Lys63), any of which can form polyubiquitin chains by linking with the carboxy-terminal glycine of another ubiquitin molecule to produce the corresponding polyubiquitination modifications. Among these seven types of polyubiquitination, only Lys48- and Lys63-linked polyubiquitination have been extensively studied[5–7]. Recently, a new type of polyubiquitin chain, the linear polyubiquitin chain, has been identified. The linear polyubiquitin chain is generated by directly linking the carboxyl group of a ubiquitin with an N-terminal methionine residue of another ubiquitin. Recent progress has revealed a few functions mediated by linear ubiquitination, including activation of signaling proteins[8–10], recruitment of regulatory proteins[11], and regulation of protein levels via the proteasome pathway[12]. However, the biological significance of linear ubiquitination remains to be further illustrated.

The linear ubiquitin chain assembly complex (LUBAC), consisting of heme-oxidized IRP2 ubiquitin ligase 1L (HOIL-1L), HOIL-1L-interacting protein (HOIP, also known as RNF31), and Shank-associated RH domain-interacting protein (SHARPIN), has been identified as the ubiquitin ligase that specifically induces linear polyubiquitination[13]. To date, only a few linearly ubiquitinated substrates have been identified. For example, linear ubiquitination of NF-kappa-B essential modulator (NEMO) promotes nuclear factor kappa-B (NF-κB) activation[8]. Linearly ubiquitinated NEMO can also bind to TNF receptor-associated factor 3 (TRAF3) during viral infection[11]. Interferon regulatory factor 3 (IRF3) linear ubiquitination at both Lys193 and either Lys313 or Lys315 activates the RLR-induced IRF3-mediated pathway of apoptosis (RIPA)[9]. Linear ubiquitination of apoptosis-associated speck-like protein containing a CARD (ASC) is required for NLR family pyrin domain-containing 3 (NLRP3) inflammasome activation[10]. LUBAC induces the ubiquitination and degradation of tripartite motif-containing 25 (TRIM25)[12]. Linear ubiquitination is also a reversible process. OTU deubiquitinase with linear linkage specificity (OTULIN, also known as Gumby) has been identified to specifically cleave linear ubiquitin chains of NF-κB[14,15]. Thus far, cellular signaling proteins regulated by LUBAC or OTULIN are still mostly unknown.

In our studies on how to improve interferon (IFN) antiviral efficacy, we want to know whether signal transducer and activator of transcription 1 (STAT1) could be modified by linear ubiquitination. If so, we speculate that linear ubiquitination could have a function in regulating IFN-induced signaling and IFN antiviral activity. The IFN family comprises three members, type-I, type-II, and type-III IFNs, which execute broad-spectrum antiviral activity[16–18]. Type-I IFN (IFN-I) activates Janus kinase (JAK)-STAT signaling through the ubiquitously expressed IFN alpha/beta receptors (IFNAR1 and IFNAR2). Upon IFN-I stimulation, STAT1 is recruited to IFNAR2 and further phosphorylated at the tyrosine 701 site by JAK1[19–23]. Phosphorylated STAT1 forms the IFN-stimulated gene factor 3 (ISGF3) complex that translocates into the nucleus to induce the expression of hundreds of antiviral IFN-stimulated genes (ISGs)[24]. Although STAT1 activation has been extensively studied, how STAT1 signaling maintains homeostasis remains to be revealed.

Here, we reveal that the transcription factor STAT1 undergoes strong linear ubiquitination modifications. Intriguingly, we demonstrate that linear ubiquitination of STAT1 blocks its activation for STAT1 signaling homeostasis. However, IFN-I treatment downregulates linear ubiquitination of STAT1 and breaks STAT1 signaling homeostasis, which promotes activation of IFN-I-induced signaling. Interestingly, viral infection can stimulate strong linear ubiquitination of STAT1 to inhibit IFN-I antiviral signaling. Thus, our study suggests that linear ubiquitination is an important mechanism controlling activation of IFN-STAT1 signaling.

## Results

**STAT1 harbors linear ubiquitination mediated by HOIP.** During the identification of the signaling regulators of the transcription factor STAT1 activation, we noticed that the ubiquitin E3 ligase RNF31, also known as HOIP, was repeatedly observed among the potential STAT1-interacting proteins in mass spectrometry analysis (Fig. 1a and Supplementary Fig. 1a). Immunoprecipitation and immunofluorescence analysis confirmed that Flag-HOIP interacts with Myc-STAT1 (Fig. 1b, c). Further studies found that deletion of the TAD domain (amino acids, 703–750) of STAT1 abolished binding of Flag-HOIP (Supplementary Fig. 1b), suggesting that HOIP could interact with the TAD domain of STAT1. Importantly, endogenous HOIP constitutively interacts with endogenous STAT1 in not only several cell lines, including 2fTGH and HEK293T (Supplementary Fig. 1c), but also primary liver, spleen, and lung tissue cells (Fig. 1d).

The constitutive binding of HOIP to STAT1 suggests that STAT1 protein could undergo linear ubiquitination modification. In line with this speculation, we observed strong linear ubiquitination of endogenous STAT1 (Fig. 1e, f), which is dependent on HOIP, since knockdown of HOIP (Supplementary Fig. 1d) dramatically inhibited linear ubiquitination of both endogenous STAT1 (Fig. 1e) and exogenously expressed STAT1 (Supplementary Fig. 1e, f). Consistently, the linear ubiquitin chain assembly complex LUBAC was able to further increase total ubiquitination (Supplementary Fig. 1g) and linear ubiquitination (Fig. 1f) of STAT1. However, the ubiquitin E3 ligase inactive mutants of HOIP (C885A and C916A) lost the ability to enhance STAT1 linear ubiquitination (Fig. 1f). To further confirm that endogenous STAT1 in intact cells harbors linear ubiquitination modification, we knocked out cellular HOIP. We found that HOIP knockout abolished linear ubiquitination of cellular STAT1 (Fig. 1g). Collectively, our findings demonstrated that cellular STAT1 undergoes HOIP-induced linear ubiquitination.

**Linear ubiquitination sustains STAT1 signaling homeostasis.** We next sought to determine the biological significance of STAT1 linear ubiquitination. STAT1 protein levels were not significantly affected by either LUBAC overexpression (Fig. 2a) or HOIP knockdown (Fig. 2b). Knockdown of HOIP also did not change the mRNA levels of STAT1 or another STAT family member STAT2 (Supplementary Fig. 2a). Interestingly, LUBAC overexpression remarkably inhibited STAT1 activation induced by IFN-I, including IFNα (Fig. 2c) and IFNβ (Supplementary Fig. 2b). Conversely, knockdown or knockout of HOIP promoted STAT1 activation (Fig. 2d and Supplementary Fig. 2c). LUBAC-mediated inhibition of STAT1 activation was dependent on the ubiquitin E3 ligase activity of HOIP (Fig. 2e). Consistent with the regulation of STAT1 activation, knockdown of HOIP significantly promoted IFN-induced ISRE promoter activity (Fig. 2f), thus enhancing both IFNα- and IFNβ-induced expression of ISG mRNAs (Fig. 2g and Supplementary Fig. 2d) and ISG proteins (Supplementary Fig. 2e). HOIP knockout promoted IFNβ-induced ISG expression (Fig. 2h). Conversely, overexpression of LUBAC inhibited IFN-I-induced expression of ISGs (Supplementary Fig. 2f). As a result, HOIP knockdown significantly promoted IFN-I-mediated antiviral activity during infection of either RNA virus VSV (Fig. 2i) or DNA virus HSV (Supplementary Fig. 2g), whereas LUBAC overexpression attenuated IFN-I antiviral activity (Fig. 2j and Supplementary Fig. 2h). Together, these findings suggest that STAT1 linear ubiquitination

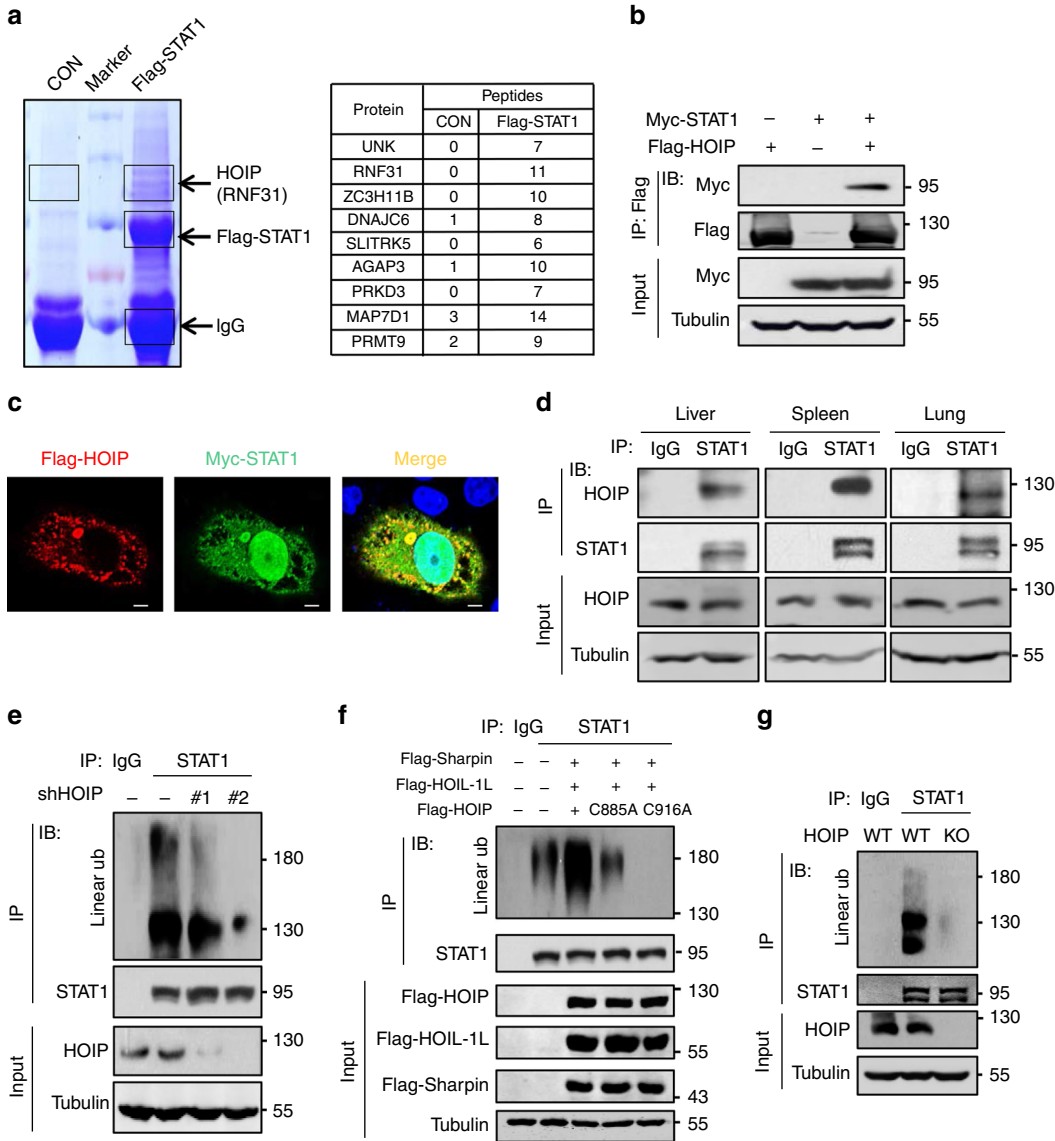

**Fig. 1 STAT1 harbors linear ubiquitination mediated by HOIP. a** Mass spectrometry analysis of the potential Flag-STAT1-binding proteins. The number of the identified peptides from the interacting proteins, including HOIP (RNF31), were shown as indicated. CON: control vectors. **b** Immunoprecipitation (IP) and immunoblotting (IB) analysis of the interaction between Flag-HOIP and Myc-STAT1 in HEK293T cells. **c** Immunofluorescence analysis of the interaction between Flag-HOIP and Myc-STAT1 in HeLa cells. Scale bar: 1 μm. **d** Immunoprecipitation analysis of the interaction between endogenous STAT1 and HOIP in mouse primary liver, spleen, and lung cells. **e** Immunoprecipitation analysis of linear ubiquitination of endogenous STAT1 in HEK293T cells transfected with either control shRNAs (−) or two different shRNAs against HOIP (shHOIP #1, #2). Ub ubiquitination. **f** Immunoprecipitation analysis of linear ubiquitination of STAT1 in HEK293T cells cotransfected with Flag-HOIL-1L and Flag-Sharpin, together with Flag-HOIP wild-type or its catalytically inactivated mutants (C885A and C916A). **g** Immunoprecipitation analysis of linear ubiquitination of endogenous STAT1 in HOIP-wild-type (WT) or HOIP-knockout (KO) HEK293T cells. Data are representative of three independent experiments.

restricts IFN-I-induced STAT1 activation and IFN-I antiviral signaling.

Our above findings have demonstrated that endogenous STAT1 in intact cells harbors strong linear ubiquitination (Fig. 1) that inhibits STAT1 activation (Fig. 2c–e). Thus, we speculated that inhibition of STAT1 linear ubiquitination could result in excessive activation of cellular STAT1 even upon signaling by autocrine IFNs. To address this hypothesis, we first determined whether linear ubiquitination affects the production of autocrine IFNs, since it has been reported that linearly ubiquitinated NEMO can bind to TRAF3 to inhibit virus-stimulated IFN production[11]. However, we observed that LUBAC overexpression did not significantly affect the production of basal levels of all three types of IFN (IFNα, IFNγ, and IFNλ) in cells without viral

infection (Fig. 2k). Intriguingly, knockdown of cellular HOIP induced substantial activation of endogenous STAT1 in cells without exogenous IFN treatment (Fig. 2l). Consistently, basal levels of ISG mRNAs were significantly upregulated by HOIP knockdown (Fig. 2m and Supplementary Fig. 2i). In line with the activation of cellular IFN-mediated antiviral signaling, the cells with HOIP knockdown showed enhanced activity to defend against virus invasion (Supplementary Fig. 2j). When cells were transfected with LUBAC, the cellular defense activity was significantly inhibited (Supplementary Fig. 2k). Importantly, in STAT1-deficient U3A cells, knockdown of HOIP lost the ability to upregulate cellular antiviral activity (Supplementary Fig. 2l), suggesting that linear ubiquitination-mediated regulation of cellular antiviral defenses depends on STAT1. Thus, given that

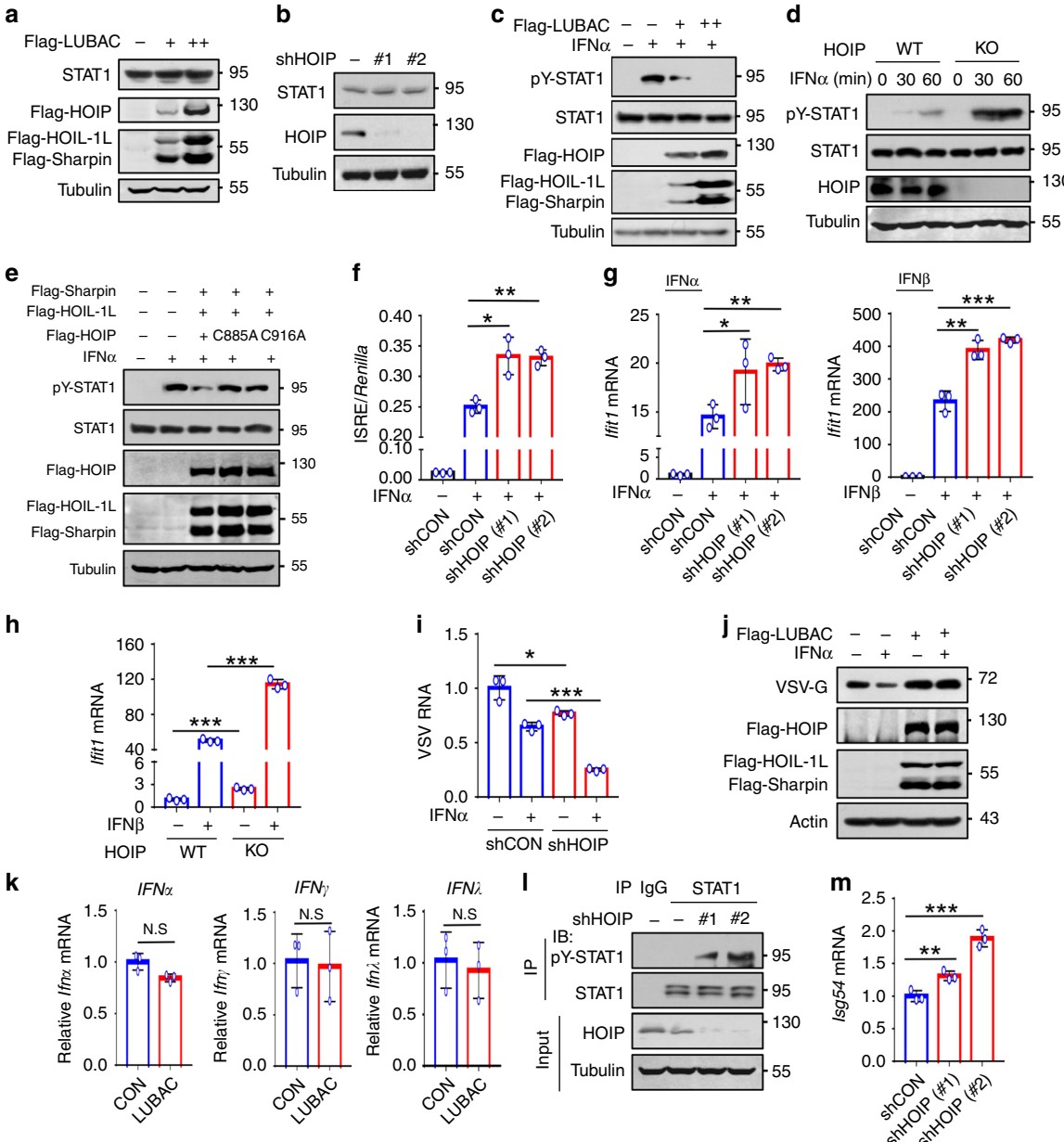

**Fig. 2 Linear ubiquitination sustains STAT1 signaling homeostasis. a** Western blot analysis of endogenous STAT1 in HEK293T cells transfected with Flag-LUBAC. **b** Western blot analysis of endogenous STAT1 in HEK293T cells transfected with control shRNAs (–) or shHOIP (#1, #2). **c** Western blot analysis of pY701-STAT1 (pY-STAT1) levels in HEK293T cells transfected with increasing amounts of Flag-LUBAC. **d** Western blot analysis of pY-STAT1 in HOIP-WT or HOIP-KO HEK293T cells treated with IFNα (1000 IU/ml). **e** Western blot analysis of pY-STAT1 in HEK293T cells cotransfected with Flag-Sharpin and Flag-HOIL-1L, together with Flag-HOIP wild-type or its catalytically inactivated mutants (C885A and C916A), and then treated with IFNα (1000 IU/ml) for 30 min. **f** Dual-luciferase reporter assay of the ISRE activity in HEK293T cells cotransfected with control shRNAs (shCON) or shHOIP (#1, #2), together with ISRE luciferase and *Renilla*, and then stimulated with IFNα (1000 IU/ml) for 20 h. **g** RT-qPCR analysis of the representative ISG (*Ifit1*) mRNA in HEK293T cells transfected with either shCON or shHOIP (#1, #2), and then stimulated with IFNα or IFNβ (1000 IU/ml) for 4 h. **h** RT-qPCR analysis of *Ifit1* mRNA in HOIP-WT or HOIP-KO HEK293T cells treated with IFNβ (1000 IU/ml) for 4 h. **i** RT-qPCR analysis of VSV viral RNA in 2fTGH cells transfected with shHOIP and then stimulated with IFNα (60 IU/ml) for 20 h, followed by infection with VSV (MOI = 0.1) for 24 h. **j** Western blot analysis of VSV-encoded proteins VSV-G in HeLa cells transfected with Flag-LUBAC and then treated as **i**. **k** RT-qPCR analysis of IFN (*Ifnα*, *Ifnγ*, and *Ifnλ*) mRNA in HEK293T cells transfected with LUBAC. **l** Immunoprecipitation analysis of pY-STAT1 in HEK293T cells transfected with shCON (–) or shHOIP (#1, #2). **m** RT-qPCR analysis of a representative ISG (*Isg54*) mRNA in HEK293T cells transfected with shHOIP (#1, #2). N.S., not significant ($p > 0.05$) and \*$p < 0.05$, \*\*$p < 0.01$, \*\*\*$p < 0.001$ (two-tailed unpaired Student's *t* test). Data are shown as mean and s.d. of three biological replicates (**f–i**, **k**, **m**), or are representative of three independent experiments (**a–e**, **j**, **l**).

inhibition of linear ubiquitination results in excessive activation of STAT1 signaling, we believe that linear ubiquitination maintains cellular IFN signaling homeostasis.

**STAT1 has linear ubiquitination at Lys511 and Lys652.** We next analyzed the linear ubiquitination-mediated regulation of different signaling proteins in the IFN-I pathway. Overexpression of LUBAC strongly upregulated linear ubiquitination of STAT1, but not Tyk2, JAK1, and STAT2 (Fig. 3a). The levels of IFNAR1 (Supplementary Fig. 3a) and IFNAR2 (Supplementary Fig. 3b, c) were not affected by either HOIP knockdown or LUBAC over-expression. In addition, linear ubiquitination did not affect IFN-I-induced activation of JAK1 and Tyk2 (Fig. 3b), as well as STAT2 (Supplementary Fig. 3d), suggesting that LUBAC could target STAT1 to regulate STAT1 activation.

We further analyzed the putative lysine residues of STAT1 with ubiquitination modifications from the PhosphoSitePlus database. We noticed that there are seven ubiquitinated lysines (Lys) on STAT1 that were identified by mass spectrometry multiple times (Supplementary Fig. 3e). We mutated these lysines (K) to arginines (R), and found that mutation of either K511 or K652 significantly reduced linear ubiquitination levels of STAT1, compared with wild-type (WT) STAT1 and the other five STAT1 mutants (Fig. 3c and Supplementary Fig. 3f). Importantly, both Lys511 and Lys652 are highly conserved in *Homo sapiens*, *Mus musculus*, *Rattus norvegicus,* and *Xenopus laevis* (Fig. 3d). Furthermore, the double mutations (DM) of STAT1 at Lys511 and Lys652 largely abolished STAT1 linear ubiquitination (Fig. 3e). In fact, the mutations at these residues of STAT1 did not inhibit the binding of HOIP (Fig. 3f). However, mutating Lys511 and Lys652 abolished the linear ubiquitination effects of LUBAC on STAT1 (Fig. 3g). Consistent with the above findings showing that inhibition of STAT1 linear ubiquitination enhanced STAT1 signaling, we found that mutating Lys511 and Lys652 significantly promoted both IFNα- and IFNβ-induced STAT1 activation (Fig. 3h and Supplementary Fig. 3g) and ISRE promoter activity (Fig. 3i). Furthermore, mutating Lys511 and Lys652 upregulated basal levels of ISGs (Supplementary Fig. 3h), promoted ISG induction by IFNβ (Fig. 3j), and improved cellular IFN-I antiviral response (Fig. 3k and Supplementary Fig. 3i). However, either LUBAC overexpression or DM mutation of STAT1 did not affect type-II IFN (IFNγ)-induced STAT1 activation (Supplementary Fig. 3j, k). Consistently, knockdown of HOIP did not promote ISG expression induced by IFNγ (Supplementary Fig. 3l). Moreover, mutating Lys511 and Lys652 did not enhance IFNγ-induced ISG expression (Supplementary Fig. 3m). Collectively, these findings demonstrated that LUBAC induces linear ubiquitination at Lys511 and Lys652 of STAT1, which inhibits IFN-I antiviral response.

**Linear ubiquitination blocks the binding of STAT1 to IFNAR2.** We further explored how STAT1 linear ubiquitination blocks its activation. We noticed that HOIP knockdown can affect cytoplasmic phosphorylated STAT1 levels (Supplementary Fig. 4a), suggesting that linear ubiquitination regulates STAT1 phosphorylation possibly before phosphorylated STAT1 enters the nucleus. In addition, mutating either Tyr701 or Ser727 of STAT1 did not inhibit HOIP binding (Supplementary Fig. 4b). The linear ubiquitination levels of STAT1-Y701F mutant were comparable with that of STAT1-WT (Supplementary Fig. 4c), indicating that Tyr701 phosphorylation of STAT1 is not required for LUBAC binding and subsequent linear ubiquitination mod-ification. Together, the results suggest that LUBAC could induce STAT1 linear ubiquitination before STAT1 gets phosphorylated.

In conjunction with the above analysis on IFN signaling proteins upstream of STAT1 (Fig. 3a, b and Supplementary Fig. 3a–c), we speculated that STAT1 linear ubiquitination could directly inhibit JAK1-induced STAT1 activation. In line with this hypothesis, we found that LUBAC overexpression dramatically inhibited STAT1 binding with JAK1 (Fig. 4a). Conversely, HOIP knockdown induced a strong interaction between STAT1 and JAK1 in both HeLa and HEK293T cells (Fig. 4b). Given that JAK1 constitutively binds to IFNAR2 that recruits STAT1 and subsequently results in STAT1 phosphorylation by JAK1, we next analyzed the direct interaction between STAT1 and IFNAR2. Consistently, knockdown of HOIP promoted the binding of STAT1 to both exogenously expressed IFNAR2 (Fig. 4c) and endogenous IFNAR2 (Fig. 4d). LUBAC overexpression robustly inhibited the binding of STAT1 to HA-IFNAR2 (Supplementary Fig. 4d) and endogenous IFNAR2 (Supplementary Fig. 4e) in cells treated with IFN-I.

Furthermore, mutation of either Lys511 or Lys652 (Supplementary Fig. 4f) or both (Fig. 4e, lane 2 vs 4) enhanced the interaction between STAT1 and IFNAR2. IFN-I treatment promoted STAT1-WT recruitment to IFNAR2 (Fig. 4e, lane 2 vs 3), whereas STAT1-DM was recruited to IFNAR2 more strongly upon IFN-I treatment (Fig. 4e, lane 3 vs 5). LUBAC overexpression inhibited the IFNAR2-STAT1 interaction, whereas the binding of IFNAR2 with STAT1-K511R or STAT1-K652R was not significantly affected by LUBAC overexpression (Supplementary Fig. 4g). Next, Flag-STAT1-WT, Myc-STAT1-K511/652R, and HA-IFNAR2 were overexpressed in a plate of STAT1-deficient U3A cells. Using Myc or Flag antibodies, we pulled down Flag-STAT1-WT and Myc-STAT1-K511/652R separately, and then observed their interaction with HA-IFNAR2. We found that Myc-STAT1-K511/652R had a stronger interaction with IFNAR2, compared with Flag-STAT1-WT (Fig. 4f), suggesting that some of the overexpressed STAT1-WT, but not STAT1-K511/652R, could undergo linear ubiquitination modifications, which inhibited their binding to HA-IFNAR2. An in vitro interaction assay also confirmed that STAT1-DM interacted more strongly with IFNAR2 (Fig. 4g). We also demonstrated that mutation of Lys511 and Lys652 did not affect the ability of STAT1 to accept Tyr701 phosphorylation, since JAK1 can normally induce in vitro Tyr701 phosphorylation of STAT1 (Fig. 4h). These findings demonstrated that linear ubiquitination-mediated inhibition of STAT1 phosphorylation is not due to the disruption of the JAK1-STAT1 phosphorylation reaction. Furthermore, cells were trans-fected with Flag-STAT1 and (or) Myc-IFNAR2 (Fig. 4i). We noticed that overexpressed Myc-IFNAR2 did not harbor noticeable linear ubiquitination (Fig. 4i, lane 2). When Myc-IFNAR2 and Flag-STAT1 were pulled down in an immunocomplex by Myc antibodies, we noticed that the Flag-STAT1 interacting with Myc-IFNAR2 did not harbor linear ubiquitination modification (Fig. 4i, lane 3). However, the Flag-STAT1 in the supernatant of Myc-IFNAR2 immunoprecipitation that does not interact with Myc-IFNAR2 had strong linear ubiquitination (Fig. 4i, lane 4), suggesting that linearly ubiquitinated STAT1 did not bind to IFNAR2. Taken together, these findings demonstrated that STAT1 linear ubiquiti-nation at Lys511 and Lys652 blocks its interaction with IFNAR2.

**IFN-I stimulation removes STAT1 linear ubiquitination.** Given that cellular STAT1 harbors linear ubiquitination that inhibits STAT1 activation, we next wondered how IFN-I stimulation is able to activate STAT1. We noticed that IFN-I-induced Tyr701-phosphorylated STAT1 did not harbor linear ubiquitination (Fig. 5a). Thus, we speculated that IFN-I signaling could remove STAT1 linear ubiquitination for STAT1 binding to IFNAR2 and subsequent activation. In line with this speculation, both IFNα and IFNβ treatment induced rapid removal of linear ubiquitin

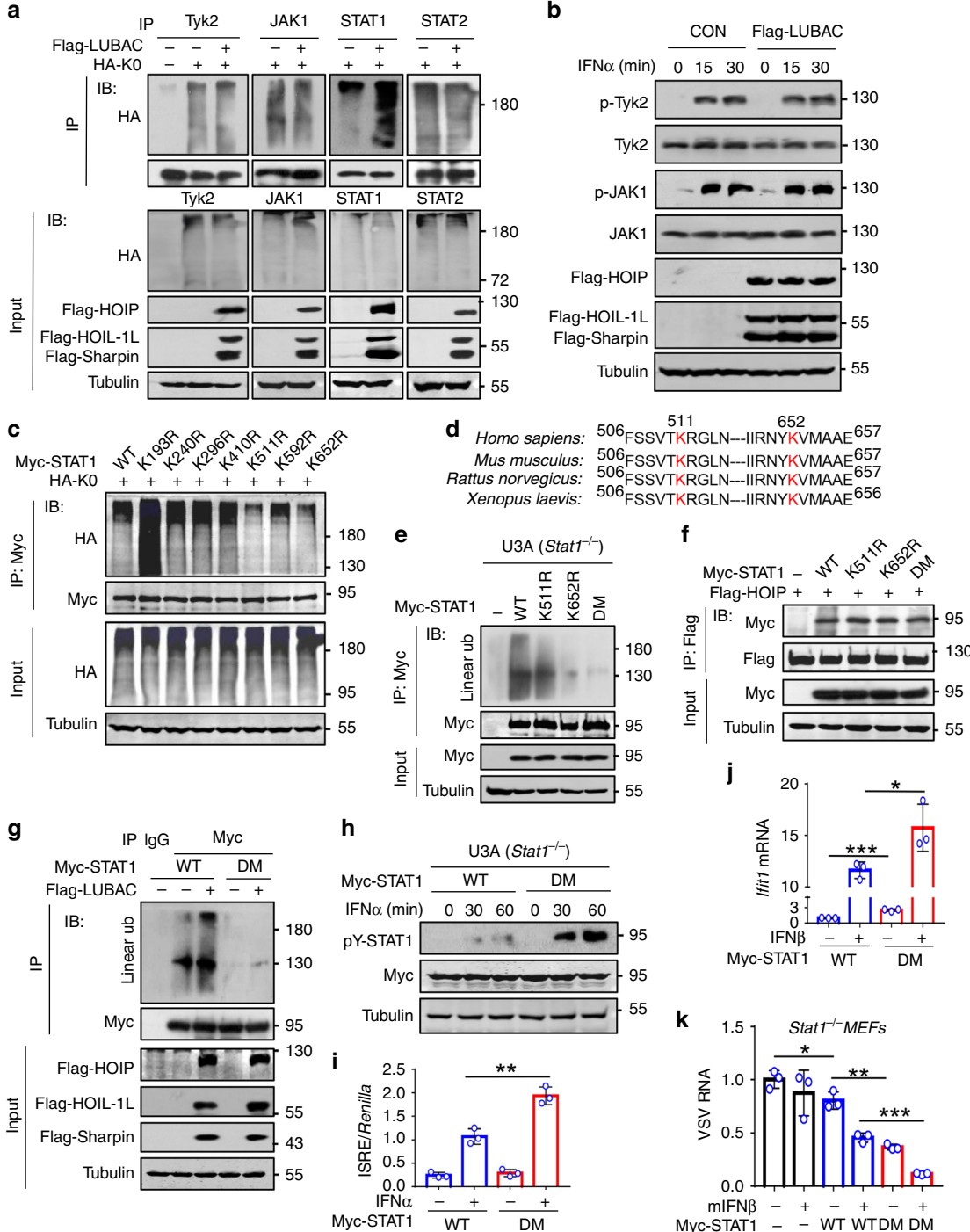

**Fig. 3 STAT1 has linear ubiquitination at Lys511 and Lys652. a** Immunoprecipitation analysis of ubiquitination of endogenous Tyk2, JAK1, STAT1, and STAT2 in HEK293T cells cotransfected with Flag-LUBAC and HA-Ub-K0 (HA-K0, all lysines on Ub are mutated to arginine) using a HA antibody. **b** Western blot analysis of the JAK-STAT signaling proteins (p-Tyk2, Tyk2, p-JAK1, and JAK1) in HEK293T cells transfected with Flag-LUBAC and then treated with IFNα (1000 IU/ml) as indicated. **c** Immunoprecipitation analysis of STAT1 ubiquitination in HEK293T cells cotransfected with Myc-STAT1 (WT or its mutants) and HA-K0. **d** Highly conserved lysine (K) residues (K511 and K652) on STAT1 from different species. **e** Immunoprecipitation analysis of linear ubiquitination of STAT1 in U3A ($Stat1^{-/-}$) cells transfected with Myc-STAT1 (WT), -K511R, -K652R, or -DM (K511/652R). **f** Immunoprecipitation analysis of the interaction between Flag-HOIP and Myc-STAT1 (WT, K511R, K652R, and DM) in HEK293T cells. **g** Immunoprecipitation analysis of linear ubiquitination of STAT1 in HEK293T cells cotransfected with Myc-STAT1 (WT or DM) and (or) Flag-LUBAC. **h** Western blot analysis of pY-STAT1 in U3A cells transfected with Myc-STAT1 (WT or DM) and then treated with IFNα (1000 IU/ml) as indicated. **i** Dual-luciferase reporter assay of the ISRE activity in HEK293T cells cotransfected with Myc-STAT1 (WT or DM), ISRE luciferase, and *Renilla*, and then stimulated with IFNα (1000 IU/ml) for 20 h. **j** RT-qPCR analysis of *Ifit1* mRNA in U3A cells transfected with Myc-STAT1 (WT or DM), and then treated with IFNβ (1000 IU/ml) for 4 h. **k** RT-qPCR analysis of VSV viral RNA in $Stat1^{-/-}$ MEF cells transfected with Myc-STAT1 (WT or DM), and then treated with mIFNβ (60 IU/ml) for 20 h, followed by infection with VSV (MOI = 0.1) for 24 h. *$p < 0.05$, **$p < 0.01$, and ***$p < 0.001$ (two-tailed unpaired Student's *t* test). Data are shown as mean and s.d. of three biological replicates (**i**, **j**, **k**), or are representative of three independent experiments (**a**–**c**, **e**–**h**).

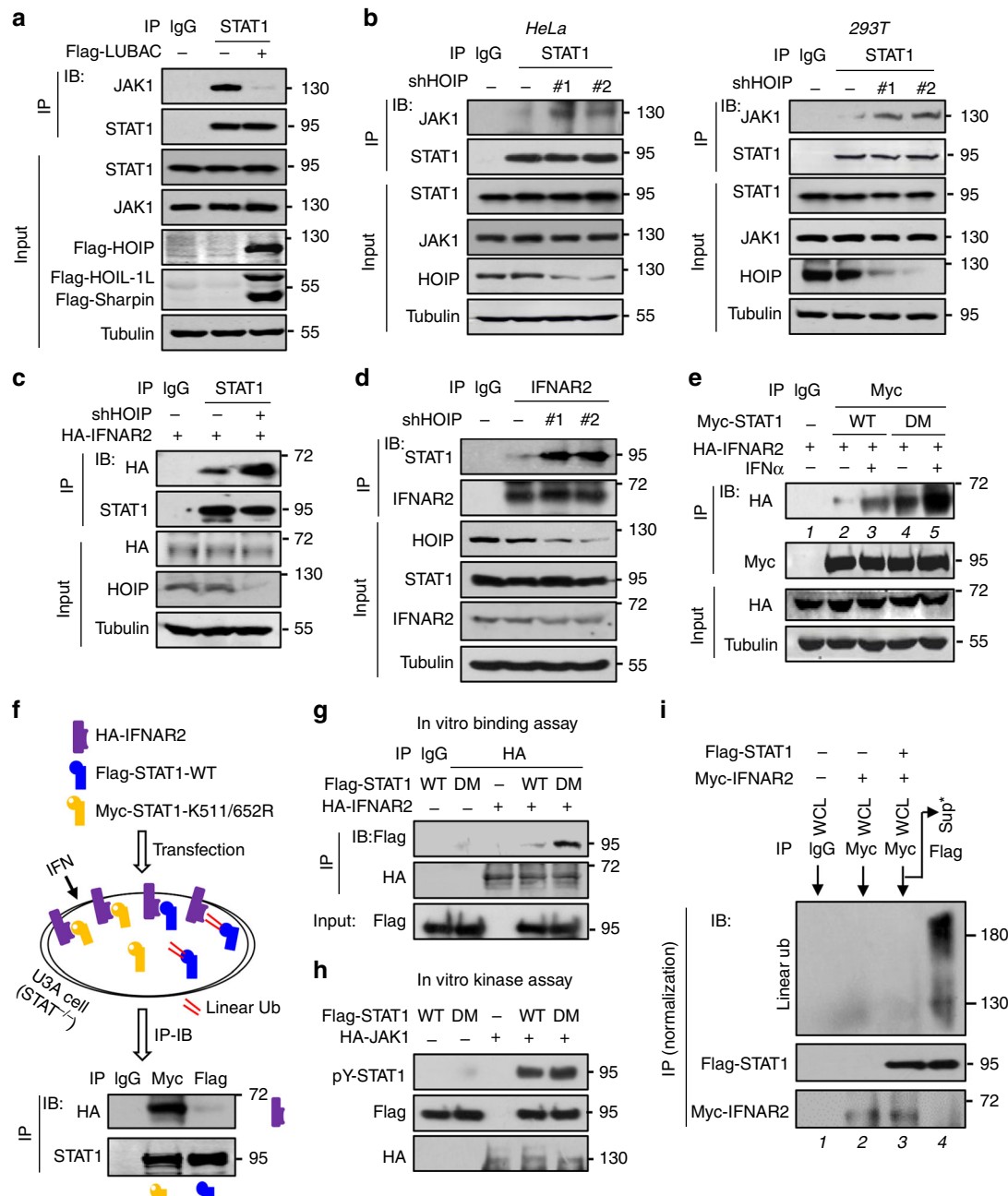

**Fig. 4 Linear ubiquitination blocks the binding of STAT1 to IFNAR2. a** Immunoprecipitation analysis of the interaction between endogenous JAK1 and STAT1 in HEK293T cells transfected with or without Flag-LUBAC and then treated with IFNα (1000 IU/ml) for 15 min. **b** Immunoprecipitation analysis of the interaction between endogenous JAK1 and STAT1 in HeLa (left) and HEK293T (right) cells transfected with shHOIP (#1, #2) and then treated with IFNα as **a**. **c** Immunoprecipitation analysis of the interaction between HA-IFNAR2 and endogenous STAT1 in HEK293T cells transfected with shHOIP and then treated with IFNα as **a**. **d** Immunoprecipitation analysis of the interaction between endogenous IFNAR2 and STAT1 in HEK293T cells treated as **b**. **e** Immunoprecipitation analysis of the interaction between HA-IFNAR2 and Myc-STAT1 in HEK293T cells cotransfected with HA-IFNAR2 and Myc-STAT1-WT or Myc-STAT1-K511/652 R (DM) and then treated with IFNα as **a**. **f** Immunoprecipitation analysis of the interaction between HA-IFNAR2 and STAT1 (WT or DM) in a plate of U3A cells cotransfected with HA-IFNAR2 (purple rectangle), Flag-STAT1-WT (blue rectangle), and Myc-STAT1-DM (orange rectangle) and then treated with IFNα as **a**. Double red lines: linear ubiquitination. **g** In vitro binding assay to analyze the interaction between HA-IFNAR2 and Flag-STAT1 (WT or DM) that were immunoprecipitated from HEK293T cells transfected with either HA-IFNAR2 or Flag-STAT1. **h** In vitro kinase assay to analyze the phosphorylation effect of HA-JAK1 on Flag-STAT1 (WT or DM). **i** HEK293T cells were transfected with Myc-IFNAR2 and (or) Flag-STAT1. The Flag-STAT1 proteins binding with Myc-IFNAR2 were immunoprecipitated by Myc antibodies. The supernatant from Myc immunoprecipitation was subjected to further immunoprecipitation using Flag antibodies to pull down those Flag-STAT1 proteins that did not interact with Myc-IFNAR2. Linear ubiquitination was analyzed as indicated. Data are representative of three independent experiments.

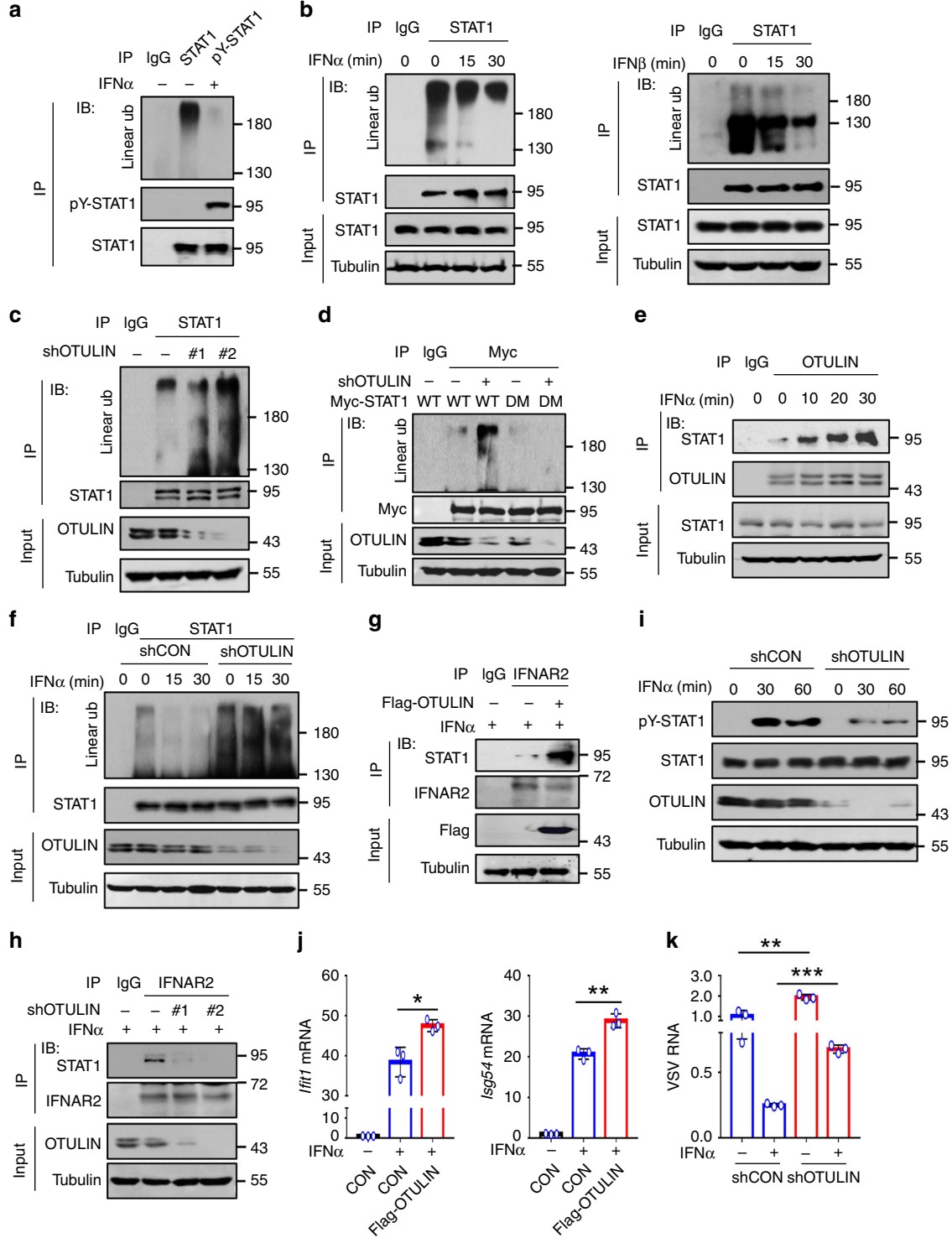

chains from STAT1 in a time-dependent manner (Fig. 5b and Supplementary Fig. 5a–c). We further observed the possible effect of OTULIN on STAT1 linear ubiquitination in IFN-I signaling. OTULIN was able to interact with cellular STAT1 (Supplementary Fig. 5d). Overexpression of OTULIN removed linear ubiquitination of cellular STAT1 (Supplementary Fig. 5e, f). Knockdown of OTULIN upregulated linear ubiquitination of cellular STAT1 (Fig. 5c). Mutation of Lys511 and Lys652 of STAT1 abolished OTULIN-mediated regulation of STAT1 linear ubiquitination (Fig. 5d), suggesting that OTULIN could be a

deubiquitinase that removes STAT1 linear ubiquitination. Interestingly, IFNα treatment significantly promoted the binding of OTULIN to STAT1 in a time-dependent manner (Fig. 5e and Supplementary Fig. 5g, h). When OTULIN was knocked down, IFN-I was no longer able to induce downregulation of STAT1 linear ubiquitination (Fig. 5f), suggesting that IFN-I-mediated removal of STAT1 linear ubiquitination is dependent on OTULIN.

Given that IFN-I signaling utilized OTULIN to remove linear ubiquitination of STAT1, we speculated that overexpression of

**Fig. 5 IFN-I stimulation removes STAT1 linear ubiquitination. a** Immunoprecipitation analysis of linear ubiquitination of STAT1 and pY-STAT1 in HEK293T cells treated with IFNα (1000 IU/ml) for 30 min. **b** Immunoprecipitation analysis of linear ubiquitination of endogenous STAT1 in 2fTGH cells treated with IFNα or IFNβ (1000 IU/ml) as indicated. **c** Immunoprecipitation analysis of linear ubiquitination of STAT1 in HEK293T cells transfected with shRNAs against OTULIN (shOTULIN, #1 and #2). **d** Immunoprecipitation analysis of linear ubiquitination of STAT1 in HEK293T cells transfected with Myc-STAT1 (WT or DM) and shOTULIN. **e** Immunoprecipitation analysis of the interaction between OTULIN and STAT1 in 2fTGH cells treated with IFNα (1000 IU/ml) as indicated. **f** Immunoprecipitation analysis of linear ubiquitination of STAT1 in HEK293T cells transfected with shOTULIN and then treated with IFNα (1000 IU/ml) as indicated. **g, h** Immunoprecipitation analysis of the interaction between endogenous IFNAR2 and STAT1 in HEK293T cells transfected with either Flag-OTULIN (**g**) or shOTULIN (#1, #2) (**h**), and treated with IFNα (1000 IU/ml) for 15 min. **i** Western blot analysis of pY-STAT1 in HEK293T cells transfected with shOTULIN and then treated with IFNα (1000 IU/ml) as indicated. **j** RT-qPCR analysis of the representative ISGs (*Ifit1* and *Isg54*) mRNA in HEK293T cells transfected with Flag-OTULIN and stimulated with IFNα (1000 IU/ml) for 4 h. **k** RT-qPCR analysis of VSV viral RNA in 2fTGH cells transfected with shOTULIN and then stimulated with IFNα (60 IU/ml) for 20 h, followed by infection with VSV (MOI = 0.1) for 24 h. $*p < 0.05$, $**p < 0.01$, and $***p < 0.001$ (two-tailed unpaired Student's $t$ test). Data are shown as mean and s.d. of three biological replicates (**j**, **k**), or are representative of three independent experiments (**a–i**).

OTULIN could promote the binding of STAT1 to IFNAR2 in IFN-I signaling. In line with this hypothesis, we observed that in IFN-I signaling, the STAT1-IFNAR2 interaction was significantly enhanced by OTULIN overexpression (Fig. 5g). Conversely, knockdown of OTULIN inhibited the binding of STAT1 to IFNAR2 in IFN-I signaling (Fig. 5h). As a result, OTULIN knockdown restricted STAT1 activation (Fig. 5i and Supplementary Fig. 5i). Consistently, OTULIN regulated IFN-I-induced ISG expression (Fig. 5j) and subsequent IFN-I-mediated antiviral activity (Fig. 5k and Supplementary Fig. 5j). Taken together, these findings demonstrated that IFN-I signaling utilizes OTULIN to downregulate linear ubiquitination of STAT1 for activation of IFN-I antiviral signaling.

**Viruses upregulate HOIP and STAT1 linear ubiquitination.** We further analyzed whether and how viral infection regulates STAT1 linear ubiquitination. We noticed that SeV infection did not affect HOIP mRNA levels within 3 h of infection (Fig. 6a). Interestingly, HOIP mRNA levels were gradually upregulated after 6 h of infection with SeV (Fig. 6a). It is well known that viral infection induces NF-κB activation and IFN production. We found that the NF-κB inhibitor PDTC inhibited HOIP mRNA upregulation during viral infection (Fig. 6b). However, IFN-I treatment did not upregulate HOIP mRNA expression (Fig. 6c). These findings suggest that virus-induced HOIP expression depends on the NF-κB pathway. Consistently, different viruses, including VSV and SeV, upregulated HOIP protein levels (Fig. 6d and Supplementary Fig. 6a). Furthermore, both VSV and SeV infections induced an enhanced interaction between HOIP and STAT1 (Fig. 6e and Supplementary Fig. 6b), thus resulting in the upregulation of STAT1 linear ubiquitination in a dose- (Fig. 6f) and a time-dependent manner (Fig. 6g and Supplementary Fig. 6c, d). When cellular HOIP was knocked out, viral infection was no longer able to upregulate STAT1 linear ubiquitination (Fig. 6h). Mutating Lys511 and Lys652 of STAT1 blocked virus-induced upregulation of STAT1 linear ubiquitination (Fig. 6i). Given that viruses rapidly induce IFN production, we further observed the dynamic regulation of STAT1 linear ubiquitination during the early stage of viral infection. The results showed that STAT1 linear ubiquitination was gradually downregulated after 1 h of viral infection, which is consistent with IFNβ induction by viral infection (Fig. 6j). However, STAT1 linear ubiquitination was strongly upregulated after 6 h of viral infection, when HOIP expression was substantially induced by viruses (Fig. 6j). Furthermore, HOIP knockout promoted STAT1 activation during viral infection (Fig. 6k). Mutating Lys511 and Lys652 of STAT1 significantly promoted ISG expression in response to viral infection (Fig. 6l). These findings demonstrated that viruses upregulate STAT1 linear ubiquitination to negatively regulate IFN-I antiviral response.

In addition, we noticed that viral infection can also upregulate linear ubiquitination of STAT1-Y701F (Supplementary Fig. 6e), which is consistent with our above analysis, demonstrating that the Tyr701 residue is not required for STAT1 linear ubiquitination. However, further mutation of Lys511 and Lys652 on the base of STAT1-Y701F abolished virus-induced linear ubiquitination of STAT1 (Supplementary Fig. 6f). Moreover, although mutation of Lys511 and Lys652 on STAT1-WT promoted virus-induced expression of ISGs, mutation of Lys511 and Lys652 on STAT1-Y701F cannot enhance virus-induced ISG expression (Supplementary Fig. 6g). Consistent with these observations, mutation of Lys511 and Lys652 on STAT1-Y701F cannot enhance cellular antiviral response (Supplementary Fig. 6h). Taken together, these findings suggested that both Lys511 and Lys652 (linear ubiquitination) and Tyr701 (phosphorylation) are involved in linear ubiquitination-mediated regulation of IFN-I response during viral infection.

**Linear ubiquitination regulates IFN-I antiviral activity.** We further explored the effect of linear ubiquitination on host IFN-I signaling and in vivo antiviral activity. The full knockout of either HOIP or HOIL-1L is lethal. Thus, HOIL-1L heterozygous ($Rbck1^{+/-}$) mice have been widely used as an effective linear ubiquitination-deficient model. We found that HOIL-1L protein levels were significantly reduced in $Rbck1^{+/-}$ mice, as compared with $Rbck1^{+/+}$ wild-type (WT) mice (Supplementary Fig. 7a). Consistently, the levels of STAT1 linear ubiquitination in the lung, spleen, liver, and heart of $Rbck1^{+/-}$ mice were much lower than those of $Rbck1^{+/+}$ mice (Fig. 7a and Supplementary Fig. 7b). To directly observe IFN-I activity in vivo, we injected mouse IFNβ (mIFNβ) into WT and $Rbck1^{+/-}$ mice. We observed that $Rbck1^{+/-}$ mice showed much stronger STAT1 activation in the lungs than WT mice (Fig. 7b). Consistent with STAT1 activation, mIFNβ-induced expression of the representative ISGs, including *Ifit1*, *Rsad2*, and *Trim25* (Fig. 7c), was significantly enhanced in the different organs of $Rbck1^{+/-}$ mice, as compared with $Rbck1^{+/+}$ WT mice (Fig. 7c, d). As a consequence, the viral loads in the organs of $Rbck1^{+/-}$ mice were much less than those of WT mice (Fig. 7e).

We noticed that although mutation of Lys511 and Lys652 of STAT1 dramatically enhanced IFN-I-induced STAT1 activation and ISG expression in $Rbck1^{+/+}$ MEF cells (Supplementary Fig. 7c, d), STAT1-K511/652 R mutants largely lost the ability to promote IFN-I response in $Rbck1^{+/-}$ MEF cells (Supplementary Fig. 7c, d), suggesting that Lys511 and Lys652 of STAT1 are involved in HOIL-1L-mediated regulation of host IFN-I antiviral response. Thus, we employed a bone marrow chimera mouse model to analyze the in vivo effects of Lys511 and Lys652 of STAT1 (Fig. 8a). The lentiviruses containing GFP-STAT1 (WT or K511/652R) were first made in HEK293T cells (Supplementary

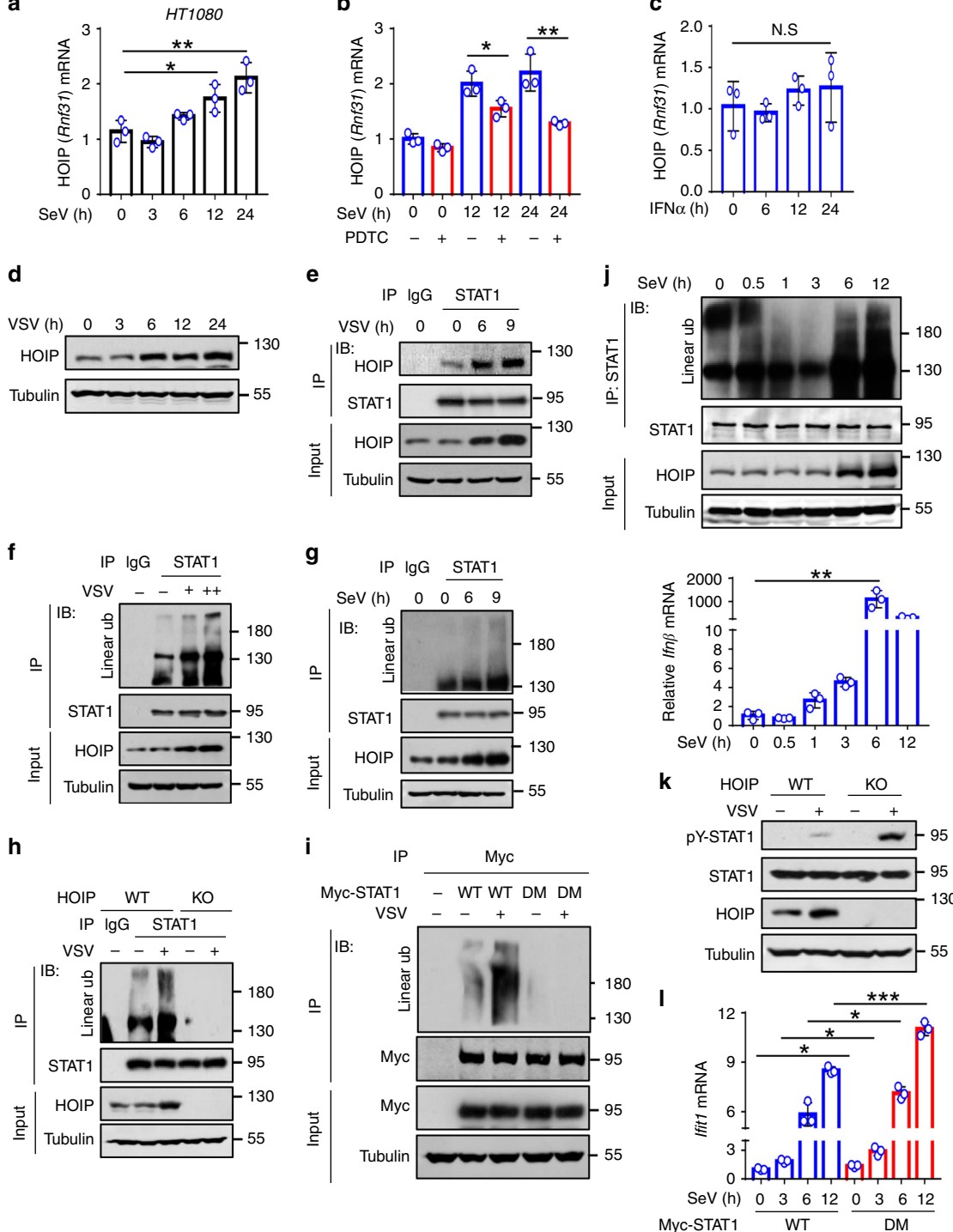

**Fig. 6 Viruses upregulate HOIP and STAT1 linear ubiquitination. a** RT-qPCR analysis of HOIP mRNA in HT1080 cells challenged with SeV (MOI = 1.0) for the indicated times. **b** RT-qPCR analysis of HOIP mRNA in HT1080 cells infected with SeV (MOI = 1.0) and then treated with a NF-κB inhibitor PDTC (10 μM) as indicated. **c** RT-qPCR analysis of HOIP mRNA in HT1080 cells stimulated with IFNα (1000 IU/ml) as indicated. **d** Western blot analysis of HOIP proteins in HEK293T cells infected with VSV (MOI = 0.1) as indicated. **e** Immunoprecipitation analysis of the interaction between HOIP and STAT1 in 2fTGH cells infected with VSV (MOI = 0.1) as indicated. **f** Immunoprecipitation analysis of linear ubiquitination of STAT1 in 2fTGH cells infected with VSV (MOI = 0.1 and 0.5) for 9 h. **g** Immunoprecipitation analysis of linear ubiquitination of STAT1 in HT1080 cells infected with SeV (MOI = 1.0) as indicated. **h** Immunoprecipitation analysis of linear ubiquitination of STAT1 in HOIP-WT or HOIP-KO HEK293T cells infected with or without VSV (MOI = 0.1) for 9 h. **i** Immunoprecipitation analysis of linear ubiquitination of STAT1 in HEK293T cells transfected with Myc-STAT1 (WT or DM) and then infected with VSV (MOI = 0.1) as indicated. **j** Immunoprecipitation analysis of linear ubiquitination of STAT1 and RT-qPCR analysis of *Ifnβ* mRNA in 2fTGH cells infected with SeV for the indicated times. **k** Western blot analysis of pY-STAT1 in HOIP-WT or HOIP-KO HEK293T cells infected with VSV (MOI = 0.1) for 12 h. **l** RT-qPCR analysis of *Ifit1* mRNA in U3A cells transfected with Myc-STAT1 (WT or DM) and then infected with SeV (MOI = 1.0) as indicated. N.S., not significant ($p > 0.05$) and $*p < 0.05$, $**p < 0.01$, $***p < 0.001$ (two-tailed unpaired Student's t test). Data are shown as mean and s.d. of three biological replicates (**a–c, l**), or are representative of three independent experiments (**d–k**).

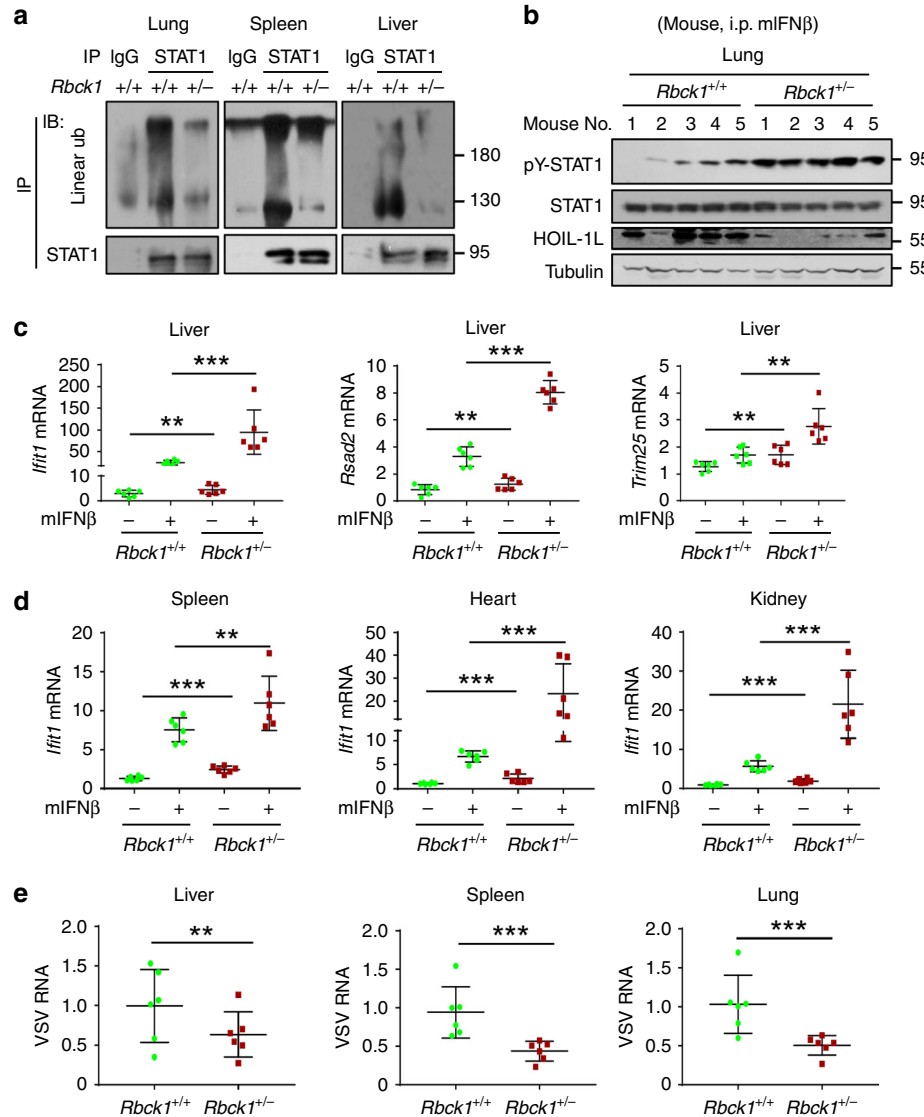

**Fig. 7 Linear ubiquitination regulates IFN-I antiviral activity. a** Immunoprecipitation analysis of linear ubiquitination of STAT1 in the lung, spleen, and liver tissues from $Rbck1^{+/+}$ or $Rbck1^{+/-}$ mice. **b** Western blot analysis of pY-STAT1 in the lung tissues from $Rbck1^{+/+}$ or $Rbck1^{+/-}$ mice administrated with mIFNβ (1500 IU/g, i.p.) for 8 h. **c** RT-qPCR analysis of the representative ISGs (*Ifit1*, *Rsad2*, and *Trim25*) mRNA in the liver tissues from $Rbck1^{+/+}$ or $Rbck1^{+/-}$ mice administrated with mIFNβ as **b. d** RT-qPCR analysis of a representative ISG (*Ifit1*) mRNA in the spleen, heart, and kidney tissues from $Rbck1^{+/+}$ or $Rbck1^{+/-}$ mice administrated with mIFNβ as **b. e** RT-qPCR analysis of VSV viral RNA in the liver, spleen, and lung tissues from $Rbck1^{+/+}$ or $Rbck1^{+/-}$ mice infected with VSV ($1 \times 10^8$ PFU per gram body mouse, i.p.) for 24 h. $**p < 0.01$ and $***p < 0.001$ (two-tailed unpaired Student's *t* test). All graphs show the mean ± SEM for six individual mice (**c–e**). Data are representative of three independent experiments (**a**, **b**).

Fig. 7e). Bone marrow cells from $Stat1^{-/-}$ donor mice were infected with these lentiviruses, and then injected into BALB/c recipient mice. These recipient mice were then administered with mIFNβ to observe IFN-I response in vivo. The results showed that mutating Lys511 and Lys652 of STAT1 resulted in enhanced IFN-I response in vivo, including IFN-I-induced STAT1-Tyr701 phosphorylation (Fig. 8b) and ISG expression (Fig. 8c). Taken together, these findings suggest that linear ubiquitination at Lys511 and Lys652 of STAT1 regulates IFN-I-mediated signaling and antiviral activity in vivo.

## Discussion

Recent identification of a distinct type of ubiquitination, linear ubiquitination, opens a new door to the study of the relationship between protein ubiquitination and signal regulation[25]. Subsequent important advances have elucidated three roles of linear ubiquitination: (1) linear ubiquitination could promote the activation of signaling proteins; (2) linear ubiquitination could result in the recruitment of specific signaling proteins; (3) linear ubiquitination could also promote protein degradation. In this study, we revealed that STAT1 harbors linear ubiquitination modifications, and linear ubiquitination of STAT1 inhibits its interaction with the membrane receptor IFNAR2, which finally blocks IFN-I-induced STAT1 activation. Thus, this study not only identified linear ubiquitination as a critical regulation of IFN-STAT1 signaling pathway, but also revealed another important role of linear ubiquitination: linear ubiquitination is able to inhibit protein–protein interaction, thereby inhibiting the activation of signaling proteins.

It is well known that the transcription factor NF-κB contains a cell-intrinsic inhibitor, IκBα. IκBα constitutively interacts with NF-κB, which prevents abnormal activation of NF-κB[26,27]. Thus, IκBα is vital for the NF-κB signaling homeostasis. Unlike NF-κB,

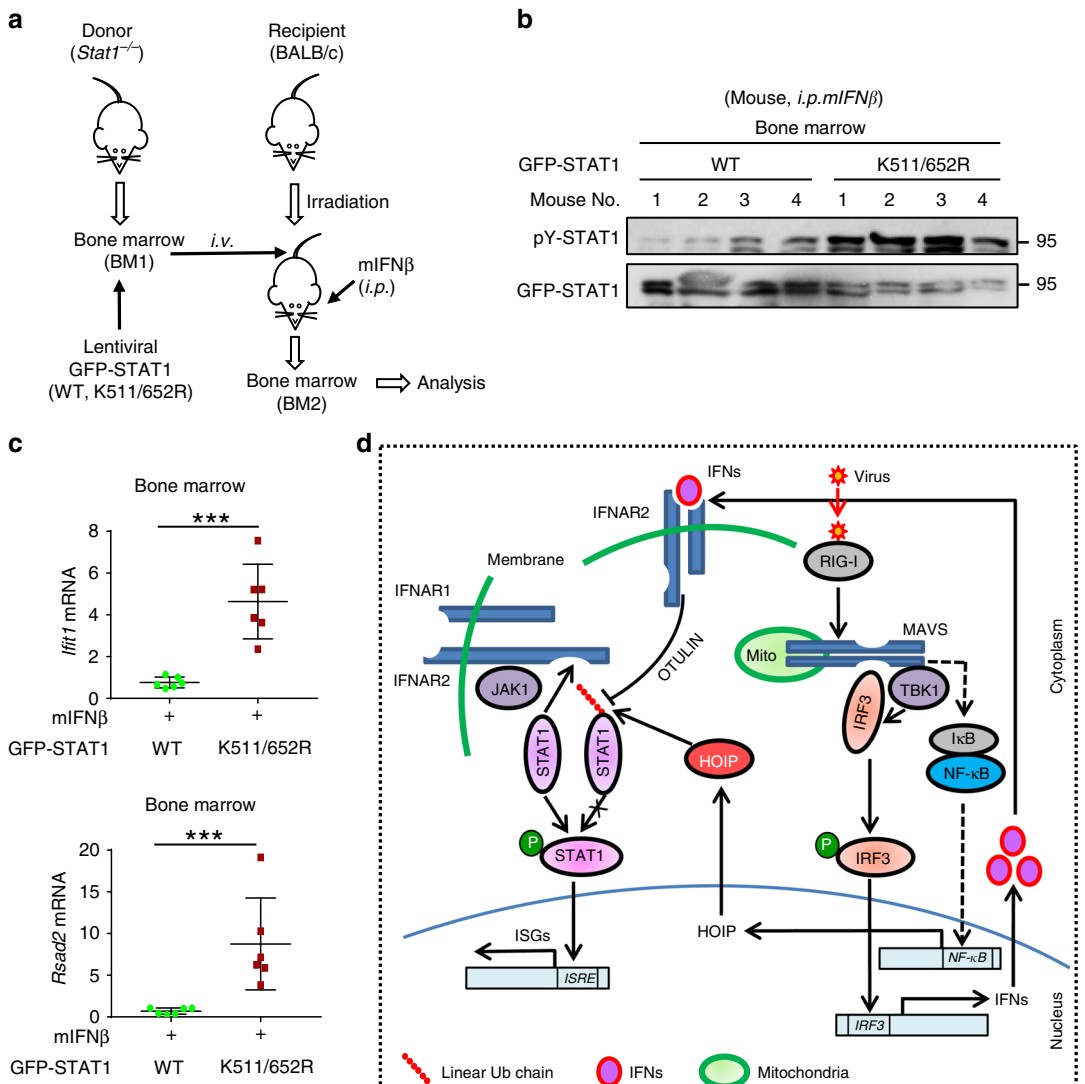

**Fig. 8 STAT1-K511/652 mutation enhances IFN-I antiviral activity. a** Diagram of bone marrow chimeras using wild-type BALB/c mice as recipients and *Stat1*⁻/⁻ (129S6) mice as donors. **b** The bone marrow was obtained from BALB/c mice that were irradiated and injected (i.v.) with the *Stat1*⁻/⁻ bone marrow with lentiviral packaging GFP-STAT1 (WT or K511/652 R), followed by administration with mIFNβ (1500 IU/g, i.p.) for 8 h. Immunoprecipitation and western blot were performed to analyze pY701-STAT1 and GFP-STAT1 levels. **c** RT-qPCR analysis of the representative ISGs (*Ifit1* and *Rsad2*) mRNA in the bone marrow from recipient mice treated as **b**. **d** Linear ubiquitination of STAT1 restricts the interaction between STAT1 and IFNAR2, thus inhibiting STAT1 activation and maintaining homeostasis of IFN signaling. Viral infection rapidly induces production of IFNs, which utilize OTULIN to remove linear ubiquitination of STAT1, thus promoting STAT1 activation. During the late stage of viral infection, HOIP expression is upregulated through the NF-κB signaling pathway, which increases STAT1 linear ubiquitination and inhibits host IFN antiviral response. ***$p < 0.001$ (two-tailed unpaired Student's *t* test). All graphs show the mean ± SEM for six individual mice (**c**). Data are representative of three independent experiments (**b**).

STAT1 does not have a cell-intrinsic inhibitor to control its activation. However, STAT1 keeps inactive in intact cells, until the stimuli induce its interaction with the signaling platform protein upstream. Interestingly, we found that in resting cells, STAT1 harbors strong linear ubiquitination modifications, which inhibit STAT1 activation. Importantly, inhibition of linear ubiquitination resulted in activated STAT1 signaling in both resting cells and an in vivo mouse model, suggesting that linear ubiquitination maintains the homeostasis of STAT1 signaling in host cells. Thus, our findings suggest that linear ubiquitination could be a common mechanism for regulation of the homeostasis of signaling proteins. In addition, recent studies have shown that HOIP or HOIL-1L knockout resulted in fetal death in mice, which was partly related to TNF-α-induced apoptosis[28,29]. Our study here suggests that abnormally activated STAT1 signaling

could also contribute to fetal death in mice with HOIP or HOIL-1L knockout. This needs to be explored further in the future.

Given that STAT1 linear ubiquitination prevents STAT1 activation, one could ask how IFN can induce STAT1 activation. To answer this question, we further investigated the relationship between IFN-I signal stimulation and STAT1 linear ubiquitination. We found that IFN-I rapidly removed linear ubiquitin chains from STAT1. OTULIN knockout inhibited down-regulation of STAT1 linear ubiquitination induced by IFN-I, which further inhibited the binding of STAT1 to IFNAR2 and subsequent STAT1 activation. Therefore, our study suggests that linear ubiquitination is similar to a lock for STAT1 signaling activation, and that OTULIN is the key. IFN-I uses OTULIN to unlock linearly ubiquitinated STAT1, allowing STAT1 to be activated. We believe that this fine regulation provides an efficient

means for the switch between the homeostasis and activation of STAT1 signaling.

Virus and host immunity are two sides of coevolution. Hosts have evolved such a perfect linear ubiquitination switch to maintain homeostasis and activate immune signals when needed. The question raised is how viruses regulate linear ubiquitination. Our study found that viral infection upregulated the mRNA and protein levels of HOIP in cells 6 h after infection. It has been reported that viral infection activates two major pathways, IFN production and NF-κB activation, which mediate cellular antiviral function[30–34]. We found that the induction of HOIP by viruses was through the NF-κB pathway. Consistent with HOIP upregulation, viral infection leads to an increase in linear ubiquitination of STAT1, which is associated with HOIP. These findings suggest that viruses utilize linear ubiquitination to inhibit the activation of STAT1 antiviral signaling, thereby restricting IFN-I innate immunity.

In summary, we revealed that the transcription factor STAT1 harbors linear ubiquitination in resting cells. Linear ubiquitination inhibits activation of STAT1, and thus maintains IFN-STAT1 signaling homeostasis. Linear ubiquitin chains of STAT1 are removed by OTULIN in IFN-I signaling, so that STAT1 is able to be activated by IFN-I stimulation. Viral infection can inhibit IFN-STAT1 antiviral signaling by stimulating HOIP expression via the NF-κB pathway and resulting in STAT1 linear ubiquitination (Fig. 8d). Our study not only reveals a new biological regulation of linear ubiquitination, but also may improve IFN-based antiviral therapy efficacy.

## Methods

**Mice.** *Rbck1*[+/−] (C57BL/6) mice were purchased from the Cyagen Biosciences Inc. (Guangzhou). *Stat1*[−/−] (129S6) mice were gifts from Dr. Y. Eugene Chin (Soochow University). Wild-type (WT) C57BL/6 mice were purchased from Shanghai SLAC Laboratory Animals. All mice were maintained under specific pathogen-free (SPF) conditions in the animal facility of Soochow University. Six- to eight-week-old mice were used in all experiments. Animal care and use protocol adhered to the National Regulations for the Administration of Affairs Concerning Experimental Animals. All animal experiments have received ethical approval by the Ethics Committee of the Soochow University, and were carried out in accordance with the Laboratory Animal Management Regulations with approval of the Scientific Investigation Board of Soochow University, Suzhou.

**Primary cells from mice.** Mouse tissues were prepared from the 6–8-week adult mice (WT or *Rbck1*[+/−]). Briefly, mouse tissues were cut into pieces and grinded to cell suspension. Mouse primary heart, liver, spleen, lung, and kidney cells were collected and prepared for further experiments.

**Cell culture and reagents.** HEK293T, A549, HeLa, 2fTGH, and HT1080 cells were obtained from ATCC. U3A cells were kindly provided by Guo-Qiang Chen (Shanghai Jiaotong University, China). Cells were cultured at 37 °C under 5% CO₂ in DMEM (HyClone) supplemented with 10% FBS (GIBCO, Life Technologies), 100 units/ml penicillin, and 100 µg/ml streptomycin. Recombinant human IFNα was purchased from PBL Interferon Source. Recombinant mouse IFNβ (mIFNβ), recombinant human IFNβ, and IFNγ were purchased from R&D Systems. Ammonium pyrrolidine dithiocarbamate (PDTC) was purchased from Selleck. Flag peptide (F3290), Puromycin, and other chemicals were purchased from Sigma.

**Plasmids and transfection.** Human Flag-HOIP, Flag-HOIL-1L, and Flag-Sharpin plasmids were gifts from Dr. Feng Shao (National Institute of Biological Sciences, China). psPAX2, pMD2G, and pCDH vectors were gifts from Dr. Qiao Cheng (Soochow University). Flag-His-OTULIN was purchased from the Vigene Biosciences. Flag-STAT1 was generated using PCR amplified from pIND-STAT1-V5 from Dr. Steven Johnson (Addgene). Myc-IFNAR2, Myc-STAT1, and the STAT1 mutants were generated using PCR methods. HA ubiquitin (HA-Ub) and HA-K0 (all lysines on the ubiquitin gene are mutated to arginines) were gifts from Dr. Lingqiang Zhang (State Key Laboratory of Proteomics, Beijing). Flag-IFNAR1, HA-IFNAR2, ISRE-Luc, and *Renilla* plasmids were gifts from Dr. Serge Y. Fuchs (University of Pennsylvania). The shOTULIN plasmid was purchased from GENECHEM (Shanghai, China). The shHOIP was constructed into the shX vector (gift from Dr. Jianfeng Dai, Soochow University).

All mutations were generated by the QuickChange Lightning site-Directed Mutagenesis Kit (Stratagene, 210518). All transient transfections were carried out

using LongTrans (Ucallm, TF/07) or GenePORTER2 (Genlantis, T202015) according to the manufacturer's instructions.

**CRISPR–Cas9-mediated genome editing.** The lenti-CRISPRv2 vector was a gift from Dr. Fangfang Zhou (Soochow University, China). Small guide RNAs were cloned into the lenti-CRISPRv2 vector, and were transfected into HEK293T cells. Forty-eight hours after transfection, the HEK293T cells were cultured under puromycin (1.5 µg/ml) selection for 2 weeks, and then the single clones were picked, grown, and identified by immunoblotting analysis. The guide RNA sequence of human *Rnf31* is 5′-GCCAGTGTGACTGTAGCAACC-3′.

**Mass spectrometry analysis.** SDS-PAGE gels were minimally stained with Silver Staining kits (Beyotime, P0017S), cut into 1 × 1-mm gel block, and digested with trypsin. The resulting tryptic peptides were purified using C18 Zip Tip. Then the peptides were analyzed by an Orbitrap Elite hybrid mass spectrometer (Thermo Fisher) coupled with a Dionex LC. MS/MS spectra were collected for the selected precursor ion within a 0.02-Da mass isolation window. Spectral data were then searched using Proteome Discoverer 1.4 against a UniProt protein database. The peptide spectrum matches (PSMs) for HOIP were obtained after database search.

**Immunoblotting and immunoprecipitation.** Cells were harvested using the lysis buffer containing 150 mM NaCl, 20 mM Tris-HCl (pH 7.4), 0.5 mM EDTA, 1% Nonidet P-40 (NP-40), PMSF (50 µg/ml), and protease inhibitor mixtures (Sigma). After centrifugation at 12 × 10³ g for 15 min, protein concentrations were measured, and equal amounts of lysates were used for immunoblotting and immunoprecipitation. Immunoprecipitation was performed using specific antibodies on a rotor at 4C. Protein G agarose beads (Millipore, #16-266) were added into samples and incubated on a rotor at 4 C. After washing five times with the lysis buffer, the immunoprecipitates were eluted by boiling with the loading buffer containing β-mercaptoethanol for 10 min, and analyzed by SDS-PAGE, followed by transferring to PVDF membranes. Membranes were then blocked with 5% nonfat milk for 30 min at room temperature, and then probed with the primary antibodies, followed by incubation with the anti-mouse or anti-rabbit (Bioworld) secondary antibodies. Immunoreactive bands were visualized with SuperSignal West Dura Extended kits (Thermo Scientific). The antibodies with the indicated dilutions were as follows: pY-STAT1 (Cell Signaling Technology, 9167, 1:1000), Linear Ubiquitin (Millipore, MABS451, 1:1000), p-JAK1 (Cell Signaling Technology, 3331S, 1:1000), p-Tyk2 (Cell Signaling Technology, 9321, 1:1000), Flag (Sigma, F7425, 1:3000), HA (Abcam, ab9110, 1:2000), JAK1 (Santa Cruz, sc-1677, 1:1000), Tyk2 (Cell Signaling Technology, 14193, 1:1000), STAT1 (Cell Signaling Technology, 9172, 1:1000), p-STAT2 (Santa Cruz, sc-21689-R, 1:500), STAT2 (Santa Cruz, sc-1668, 1:500), Myc (Abmart, M20002H, 1:3000), VSV-G (Santa Cruz, sc-66180, 1:1000), β-Actin (Proteintech, 66009-1-Ig, 1:2000), Tubulin (Proteintech, 66031-1-Ig, 1:3000), RBCK1 (SIGMA, M11547, 1:1000), Lamb1 (Proteintech, 12987-1-AP, 1:2000), Ubiquitin (Ub) (Santa Cruz, 12987-1-AP, 1:1000), OTULIN (Cell Signaling Technology, 14127, 1:2000), IFN-α/βRβ (D-6) (Santa Cruz, sc-271105, 1:500), STAT1 (Santa Cruz, sc-464, 1:1000), IFIT1 (Cell Signaling Technology, 14769, 1:1000), HOIP (Abcam, ab125189, 1:2000), and PKR (Cell Signaling Technology, 3072, 1:1000).

When protein ubiquitination was determined, cells were harvested in the RIPA lysis buffer containing *N*-ethylmaleimide (10 mM), and the immunoprecipitates were washed three times with the high-salt (500 mM NaCl) washing buffer and twice with the normal (150 mM NaCl) washing buffer. Uncropped images of all gels and blots are presented in Supplementary Fig. 8.

**RNA isolation, quantitative real-time PCR, and primers.** Total RNAs were isolated from different cell lines or mouse tissues using TRIzol reagent (Invitrogen). The cDNA was synthesized from 1 µg of total RNA using the RevertAid First Strand cDNA Synthesis kit (Thermo #K1622) and subjected to quantitative real-time PCR (RT-qPCR) with different primers in the presence of SYBR Green Supermix (BIO-RAD) using a StepOne Plus real-time PCR system (Applied Bioscience).

The relative expression of the target genes was normalized to *β-actin* mRNA. The results were analyzed from three independent experiments, and shown as the average mean ± standard deviation (s.d.).

A list of all primer sequences:
human *Rnf31*:
Forward: 5′-CTCGGTACTGGCGTGGTGTCAA-3′
Reverse: 5′-TGGTGCTCATCTGGCTCCTCCT-3′
human *Ifit1*:
Forward: 5′-CACAAGCCATTTTCTTTGCT-3′
Reverse: 5′-ACTTGGCTGCATATCGAAAG-3′
human *Isg15*:
Forward: 5′-GGGACCTGACGGTGAAGATG-3′
Reverse: 5′-CGCCGATCTTCTGGGTGAT-3′
human *Isg54*:
Forward: 5′-CACCTCTGGACTGGCAATAGC-3′
Reverse: 5′-GTCAGGATTCAGCCGAATGG-3′
human *Mx1*:
Forward: 5′-ACATCCAGAGGCAGGAGACAATC-3′

Reverse: 5′-TCCACCAGATCAGGCTTCGTCAA-3′
human *Stat1*:
Forward: 5′-ACTCAAAATTCCTGGAGCAG-3′
Reverse: 5′-ACGCTTGCTTTTCCTTATGTT-3′
human *Stat2*:
Forward: 5′-ATGCTGCAGAATCTTGACA-3′
Reverse: 5′-TAGTTCAGCTGATCCAAGAAG-3′
VSV:
Forward: 5′-ACGGCGTACTTCCAGATGG-3′
Reverse: 5′-CTCGGTTCAAGATCCAGGT-3′
SeV:
Forward: 5′-GATGACGATGCCGCAGCAGTAG-3′
Reverse: 5′-CCTCCGATGTCAGTTGGTTCACTC-3′
HSV-*Ul46*:
Forward: 5′-CTTGCCGGTCTGCCACAG-3′
Reverse: 5′-CTTGCCGGTCTGCCACAG-3′
HSV-*Icp27*:
Forward: 5′-ATCGCACCTTCTCTGTGGTC-3′
Reverse: 5′-GCAAATCTTCTGGGGTTTCA-3′
human *Ifna*:
Forward: 5′-TGGGAACAGAGCCTCCTAGA-3′
Reverse: 5′-CAGGCACAAGGGCTGTATTT-3′
human *Ifnβ*:
Forward: 5′-CATTACCTGAAGGCCAAGGA-3′
Reverse: 5′-CAGCATCTGCTGGTTGAAGA-3′
human *Ifnγ*:
Forward: 5′-TCCCATGGGTTGTGTGTTTA-3′
Reverse: 5′-AAGCACCAGGCATGAAATCT-3′
human *Ifnλ*:
Forward: 5′-CTGCTGAAGGACTGCAAGTG-3′
Reverse: 5′-GAGGATATGGTGCAGGGTGT-3′
mouse *Ifit1*:
Forward: 5′-GCCTATCGCCAAGATTTAGATGA-3′
Reverse: 5′-TTCTGGATTTAACCGGACAGCA-3′
mouse *Trim25*:
Forward: 5′-ATGGCTCAGGTAACAAGGGAG-3′
Reverse: 5′-GGGAGCAACAGGGGTTTTCTT-3′
mouse *Rsad2*:
Forward: 5′-CAGGCTGGTTTGGAGAAGATCAAC-3′
Reverse: 5′-TACTCCCCATAGTCCTTGAACCATC-3′
*β-actin*:
Forward: 5′-ACCAACTGGGACGACATGGAGAAA-3′
Reverse: 5′-ATAGCACAGCCTGGATAGCAACG-3′
shHOIP(#1):
Forward: 5′-GATCCGCTCTGAACATCCTGGAGAATTCAAGAGATTCT
CCAGGATG TTCAGAGTTTTTTG-3′
Reverse: 5′-AATTCAAAAAACTCTGAACATCCTGGAGAATCTCTTGAA
TTCTCCAGGATGTTCAGAGCG-3′
shHOIP(#2):
Forward: 5′-GATCCGAGTCAAGTCTGGTACTGTTTCAAGAGAACAGTA
CCAGAC TTGACTCTTTTTTG-3′
Reverse: 5′-
AATTCAAAAAAGAGTCAAGTCTGGTACTGTTCTCTTGAAACAGTA
CCAGACTTGACTCG-3′

**Reporter gene assay**. Cells were transfected with the ISRE luciferase and *Renilla* plasmids, together with the specific constructs (Myc-STAT1-WT or -DM, or shHOIP). Forty-eight hours after transfection, cells were treated with IFNα for 20 h. The luciferase activity was detected by the Dual-luciferase Reporter Assay System (Promega, #E1910) according to the manufacturer's protocol. Activity was assayed from three independent experiments, and shown as the average mean and standard derivation (s.d.).

**Cytoplasmic and nuclear protein extraction**. Cells were scraped in cold PBS and then harvested in lysis buffer containing 10 mM HEPES (pH 7.9), 50 mM NaCl, 0.5 mM Sucrose, 0.1 mM EDTA, 0.5% Triton X-100, 1 mM DTT, 10 mM Sodium pyrophosphate decahydrate, 0.5 M NaF, 0.2 M Na$_3$VO$_4$, 1 mM PMSF, and protease inhibitor mixtures (Sigma). Then the supernatant was collected for the cytoplasmic extract after centrifugation for 10 min at 1000 rpm. The pellet was resuspended with Buffer A containing 10 mM HEPES (pH 7.9), 10 mM KCl, 0.1 mM EGTA, 0.1 mM EDTA, 1 mM DTT, 1 mM PMSF, and protease inhibitor mixtures, followed by centrifugation for 5 min at 1000 rpm in swinging-bucket rotor. The supernatant was removed. The four volumes of buffer C containing 10 mM HEPES (pH 7.9), 500 mM NaCl, 0.1 mM EGTA, 0.1 mM EDTA, 0.1% Nonidet P-40, 1 mM DTT, 1 mM PMSF, and protease inhibitor mixtures was added, followed by the vortex (15 min at 4 °C) and centrifugation (10 min at 14,000 rpm). The supernatant was collected for the nuclear extract. Alpha-Tubulin and Lamin B1 were used as a loading control.

**Immunofluorescence microscopy**. For colocalization analysis, HeLa cells were transfected with Flag-HOIP and Myc-STAT1. Forty-eight hours after transfection, cells were washed by 1× PBS three times, fixed in 4% paraformaldehyde on ice, and then permeabilized with 0.5% Triton X-100 and blocked with 5% BSA. The cells were incubated overnight with an anti-Flag antibody and an anti-Myc antibody in 0.5% BSA. After washing three times with 1× PBS, cells were stained with 488 goat anti-mouse IgG (Alexa Fluor, A11001) or 594 goat anti-rabbit IgG (Alexa Fluor, A11012). Cell nuclei were stained with DAPI. The fluorescent images were captured with the Nikon A1 confocal microscope.

**Viruses and viral infection in vitro**. Vesicular stomatitis virus (VSV) and Sendai virus (SeV) were gifts from Dr. Chen Wang (Shanghai Institute of Biochemistry and Cell Biology, Chinese Academy of Sciences). Herpes Simplex Virus-1 (HSV-1) was from Dr. Chunfu Zheng (Fujian Medical University, China). VSV-GFP was from Dr. Chunsheng Dong (Soochow University, China). The antiviral effect of IFNα was determined by pretreating cells with IFNα overnight prior to infection with viruses. Briefly, cells were transfected with Flag-LUBAC/OTULIN constructs or HOIP/OTULIN shRNAs. Forty-eight hours after transfection, cells were pretreated with IFNα (60 IU/ml) overnight. After washing twice, cells were challenged by VSV-GFP or HSV-1 at a multiplicity of infection (MOI) of 0.1 or 1.0 in the serum-free medium for 2 h for virus entry. Then the infection medium was removed by washing twice by 1× PBS. Cells were fed with in the fresh 10% FBS medium for 24 h. Then cells were analyzed by immunofluorescence, RT-qPCR, or western blot using a VSV-G antibody. To assess the antiviral ability of cells against HSV-1, cells were collected, and viral RNAs were analyzed to detect viral *Ul46* and *Icp27* genes by RT-qPCR.

**Viral infection in vivo**. For in vivo viral infection studies, 8-week-old mice (WT or *Rbck1*$^{+/-}$) were given intraperitoneal injections (i.p.) of VSV ($1 \times 10^8$ PFU per gram body mouse). Twenty-four hours after infection, mouse lung, liver, and spleen were harvested, and RT-qPCR was used for the analysis of VSV viral RNAs.

**In vitro kinase assay**. In vitro phosphorylation of Flag-STAT1 by HA-JAK1 (via immunopurification) was carried out in the kinase buffer (50 mM Tris-HCl, pH 7.4, 10 mM MgCl$_2$, and 2 mM DTT) with 0.2 mM ATP for 30 min at 30 °C. The products of this reaction were separated by SDS-PAGE, and analyzed by immunoblotting using an anti-pY701-STAT1 antibody.

**In vitro binding assay**. HEK293T cells were transfected with the constructs. Flag-STAT1 (WT or DM) was immunoprecipitated and then eluted with the Flag peptides, respectively. HA-IFNAR2 was pulled down by the HA antibody from cells transfected with HA-IFNAR2. Then Flag-STAT1 (WT or DM) eluates were mixed and vibrated with the HA-IFNAR2 immunoprecipitates at 4 °C. The products of this reaction were separated by SDS-PAGE and analyzed by immunoblotting using the anti-Flag or anti-HA antibody.

**Interferon-β administration in vivo**. For in vivo ISG studies, 8-week-old mice (WT or *Rbck1*$^{+/-}$) were given intraperitoneal injections (i.p.) of IFNβ (1500 IU/g mouse) for 8 h. Then mouse liver tissues were harvested and analyzed for *Ifit1*, *Rsad2*, and *Trim25* mRNA levels by RT-qPCR. In addition, mouse heart, spleen, and kidney were also harvested for the analysis of the representative ISG (*Ifit1*).

**Bone marrow chimeras**. Recipient mice (wild type, BALB/c) received irradiation of 650 cGy by X-ray. Donor mice (*Stat1*$^{-/-}$, 129S6) were sacrificed to harvest bone marrow (BM) cells. *Stat1*$^{-/-}$ bone marrow cells were infected with the lentiviruses containing GFP-STAT1-WT or GFP-STAT1-K511/652R (DM) that were produced by the psPAX2, pMD2G, and pCDH package system. Then $1 \times 10^7$ bone marrow cells were injected (i.v.) into BALB/c recipient mice. After 10 days, these mice were administered with mIFNβ (1500 IU/g, i.p.) for 8 h.

**Statistical analysis**. Two-tailed unpaired Student's *t* test was used for the comparison between different groups. All differences were considered statistically significant with $p < 0.05$. *P* values are indicated by asterisks in the figures as follows: *$p < 0.05$, **$p < 0.01$, and ***$p < 0.001$.

**Reporting summary**. Further information on research design is available in the Nature Research Reporting Summary linked to this article.

## Data availability
The authors declare that all data supporting the findings of this study are available in the article and its Supplementary Information Files, or on request from the corresponding author. Raw data and the full microscopy image data sets corresponding to Fig. 1c and Supplementary Figs. 2h and 7e are supplied in the Source Data file.

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

## Acknowledgements

We thank Dr. Feng Shao (National Institute of Biological Sciences, Beijing), Dr. Chen Wang (Shanghai Institute of Biochemistry and Cell Biology, Chinese Academy of Sciences), and Dr. Lingqiang Zhang (State Key Laboratory of Proteomics, Beijing) for important reagents. We also thank Dr. Qiao Cheng (Institute of Blood and Marrow Transplantation, Soochow University) for the direction of bone marrow chimera experiments. This work is supported by grants from the National Natural Science Foundation of China (31770177 and 31970846), the National Key R&D Program of China (2018YFC1705500): 2018YFC1705505, the Priority Academic Program Development of Jiangsu Higher Education Institutions (PAPD), the program of 1000 Young Talents (2014), Jiangsu Provincial Distinguished Young Scholars (BK20130004), and the Postgraduate Research & Practice Innovation Program of Jiangsu Province (KYCX19_1980).

## Author contributions

Y.Z., Q.F., J.L., F.H., Y.M., J.L., Y.X., X.C., and H.Z. performed the experiments. J.L. and Y.M. assisted with mouse experiments, tissue processing, and analysis. T.G., Y.Y., L.Z., and J.W. assisted with the transfections, lentivirus, and RT-qPCR analysis. H.Z. and Y.Z. designed experiments, analyzed data, and wrote the paper. H.Z. and Y.Z. discussed the paper. H.Z. was responsible for research supervision, coordination, and strategy.

## Competing interests

The authors declare no competing interests.
