## [Peer Review File · Nature Communications]

Reviewers' comments:

Reviewer #1 (Remarks to the Author):

In this manuscript, the authors reported that the linear ubiquitin chain assembly complex (LUBAC) ubiquitinates STAT1 in resting cells. Linear ubiquitination of STAT1 blocks the recruitment of STAT1 to IFNAR2 and subsequent phosphorylation by JAK1. Upon IFN-IFNAR2 engagement, OTULIN removes the linear ubiquitination of STAT1 to promote its activation. Furthermore, prototype RNA viruses (Sendai virus and VSV) can induce the expression of HOIP to promote the linear ubiquitination of STAT1. Overall, this discovery is very interesting and informative in the regulation of STAT-mediated immune response. However, there are a few questions that the authors may address.

Major issues:

1. Scientifically, the manuscript is very solid and findings are supported by extensive experiments. Regarding the implication of linear ubiquitination of STAT1 in viral immune evasion, their findings did not support the conclusion. In order to support the author's conclusion, one may have to show that viruses, such as SeV and VSV, deploy specific strategies to induce this activity. Rather, it is conceivable that the virus-induced linear ubiquitination of STAT1 is a negative feedback mechanism that cells developed to avoid overt inflammation and subsequent collateral damage.
2. IFN γ , in addition to type I interferons, can induce STAT1 phosphorylation at Tyr701 that drives the homodimerization of STAT1. In the nucleus, STAT1 homodimers bind to GAS elements and promote ISG transcription. It will be interesting and meaningful to detect whether linear ubiquitination can block the activation of STAT1 in type II interferon signaling.

Minor issues:

1. Many co-IP experiments do not have input, including those shown in Figure 1d, 1g, 3f, 5e, 6e, 6f, 6g, 6h, etc.
2. When detecting the phosphorylation of STAT1, the levels of total STAT1 should also be showed.
3. In Figure 4a and 4b, the protein levels of JAK1 and STAT1 should be showed in input.
4. In Figure 4d, the protein levels of STAT1 and IFNAR2 should be showed in input.
5. In general, the manuscript is well-written. However, I would recommend proof-reading by a professional or native English-speaking individual. Some examples from the manuscript: Study the effect...to a large extent... the sentence in line 194-195... Line 215-216,... in conjugation with (line 217)(In conjunction with?)...

Reviewer #2 (Remarks to the Author):

In the manuscript by Zuo et al, the authors demonstrate a novel biochemical mechanism that negatively regulates STAT1 antiviral activity. Using extensive biochemical and genetic tools, the authors demonstrate that STAT1 is linearly ubiquitinated on at least two lysine residues by LUBAC. The linear ub of STAT1 inhibits its ability to signal by IFN- α , an antiviral cytokine. As a part of the mechanism, the linear ub of STAT1 prevents its interaction with IFNAR2 to inhibit IFN signaling. Furthermore, the authors also report that the deubiquitinase, Otulin removes the linear ub chains from STAT1 to restore the IFN signaling. Finally, the authors utilize Hoil-1 \pm mice to study the biological significance of their study, and these mice exhibit reduced viral replication presumably due to the enhanced IFN-STAT1 signaling.

The study is very interesting, and using highly sophisticated biochemical approaches, the authors clearly established a novel posttranslational modification on STAT1 to regulate the IFN signaling. IFN is an antiviral cytokine; however, excessive IFN signaling can lead to undesired outcomes including autoimmune reactions and diseased states. Therefore, this study uncovers a new mechanism that may help develop new therapy based on the linearly ubiquitinated STAT1.

Although the mechanistic part of the study is quite solid, the in vivo studies are relatively weaker. The authors should consider the following points to improve their study:

1. The lack of a STAT1 knock-in mouse harboring mutated lysines does not allow the authors to directly test the new mechanism on the biological functions of IFN. Therefore, the phenotype in Hoil-1+/- mice may not be completely due to this new mechanism. The authors should find a way to establish this in vivo to significantly strengthen their study. Is it possible to address this partly ex vivo in the KO cells, isolated from Hoil-1-/- and/or Hoip-/- cells, by expressing STAT1 Wt or the lysine mutants. Also, the authors claim that Hoil-1-/- are not available; however, Tokunaga et al (2009) reported these mice.
2. The antiviral functions of the lysine mutants should be tested in the context of IFN-treated cells in Suppl Fig 3F. These results directly test the functional contribution of STAT1 linear ub, and should be moved to the main figure.
3. Also, the antiviral functions should be tested in STAT1-/- mouse cells expressing the STAT1 mutants, to generalize the effects.
4. Virus infection induces IFN-beta in all cell types whereas IFN-alpha expression is restricted primarily to the myeloid cells. It would be relevant to also show that the new mechanism is valid in IFN-beta treated cells as well to broaden the significance of the study.
5. Many viruses (including SeV) inhibit phosphorylation of STAT1 and, therefore, it would be interesting to see whether an unphosphorylated STAT1 mutant can undergo the new regulation. This is to support the claim that viruses may use this mechanism to evade IFN responses, as well as the transcriptional functions of unphosphorylated STAT1.
6. Please provide a summary model for the study.
7. Minor point: why STAT1 appears as a doublet in some of the panels?

Dear reviewers,

Thank you so much for your very valuable comments, which are very helpful for us to improve our study. Here, we have addressed all concerns from two reviewers by carrying out a series of experiments and modifying this manuscript according to the reviewers' suggestions. Again, I greatly appreciate!

Best wishes,

Hui

Reviewers' comments

Reviewer #1 (Remarks to the Author):

In this manuscript, the authors reported that the linear ubiquitin chain assembly complex (LUBAC) ubiquitinates STAT1 in resting cells. Linear ubiquitination of STAT1 blocks the recruitment of STAT1 to IFNAR2 and subsequent phosphorylation by JAK1. Upon IFN-IFNAR2 engagement, OTULIN removes the linear ubiquitination of STAT1 to promote its activation. Furthermore, prototype RNA viruses (Sendai virus and VSV) can induce the expression of HOIP to promote the linear ubiquitination of STAT1. Overall, this discovery is very interesting and informative in the regulation of STAT-mediated immune response. However, there are a few questions that the authors may address.

Thank you so much for these good comments!

Major issues:

1. Scientifically, the manuscript is very solid and findings are supported by extensive experiments. Regarding the implication of linear ubiquitination of STAT1 in viral immune evasion, their findings did not support the conclusion. In order to support the author's conclusion, one may have to show that viruses, such as SeV and VSV, deploy specific strategies to induce this activity. Rather, it is conceivable that the virus-induced linear ubiquitination of STAT1 is a negative feedback mechanism that cells developed to avoid overt inflammation

and subsequent collateral damage.

This is a very good comment! We absolutely agree with the reviewer about the difference between viral evasion and host negative feedback. Thus, we corrected our description and conclusion in the “viral evasion” section to reflect the actual negative feedback mechanism of STAT1 linear ubiquitination induced by viral infection. Please see Line 306-335.

In addition, to better demonstrate the negative feedback mechanism, we added several experiments to observe STAT1 linear ubiquitination and ISGs induction in cells with STAT1-WT or STAT1-DM mutants under conditions of viral infection. These new results further demonstrated that virus-induced STAT1 linear ubiquitination at Lys511/652 negatively regulates host IFN response. Please see the new Fig. 6i, 6l and Fig. S6f, S6g, S6h.

2. IFN γ , in addition to type I interferons, can induce STAT1 phosphorylation at Tyr701 that drives the homodimerization of STAT1. In the nucleus, STAT1 homodimers bind to GAS elements and promote ISG transcription. It will be interesting and meaningful to detect whether linear ubiquitination can block the activation of STAT1 in type II interferon signaling.

Thanks! According to the reviewer’s suggestion, we observed the possible roles of STAT1 linear ubiquitination in IFN γ -induced STAT1 activation and ISGs expression. We noticed that overexpression of LUBAC did not noticeably inhibit IFN γ -induced STAT1 Tyr701 phosphorylation (Please see the new Fig. S3j). Furthermore, mutation of STAT1-Lys511/652 did not significantly affect STAT1 activation in STAT1-deficient U3A cells, as compared with STAT1-WT (Please see the new Fig. S3k). Moreover, both knockdown of HOIP (new Fig. S3l) and mutation of STAT1-Lys511/652 (new Fig. S3m) did not noticeably affect

IFN γ -induced ISGs expression. These results suggested a possible difference in delicate regulation of type-I and type-II IFNs signaling.

Minor issues:

1. Many co-IP experiments do not have input, including those shown in Figure 1d, 1g, 3f, 5e, 6e, 6f, 6g, 6h, etc.

Thank the reviewer for pointing out the issues. According to the reviewer's suggestion, we carefully checked the whole Figures and Suppl Figures, and added all input bands in our revised manuscript, including the new Fig. 1d, 1g, 3e, 5b, 5e, 6e, 6f, 6g, 6h, S1c, S4c, S5c, S5h, S6b, and S6d.

2. When detecting the phosphorylation of STAT1, the levels of total STAT1 should also be showed.

We have added the total STAT1 levels in the new Fig. 2c-e, 5i, 6k, 7b, S2c, S3d, S4a and S5i.

3. In Figure 4a and 4b, the protein levels of JAK1 and STAT1 should be showed in input.

We have added the protein levels of JAK1 and STAT1 as the input in Fig. 4a and 4b.

4. In Figure 4d, the protein levels of STAT1 and IFNAR2 should be showed in input.

Thanks! We have added the protein levels of STAT1 and IFNAR2 as the input in Fig. 4d.

5. *In general, the manuscript is well-written. However, I would recommend proof-reading by a professional or native English-speaking individual. Some examples from the manuscript: Study the effect...to a large extent... the sentence in line 194-195... Line 215-216,... in conjugation with (line 217)(In conjunction with?)...*

We thank the reviewer for pointing out the writing issue. We have corrected them and this manuscript have been edited by a native English-speaking editor in the American Journal Experts (the verification code: 0ACB-5065-14FA-8258-5619).

Reviewer #2 (Remarks to the Author):

In the manuscript by Zuo et al, the authors demonstrate a novel biochemical mechanism that negatively regulates STAT1 antiviral activity. Using extensive biochemical and genetic tools, the authors demonstrate that STAT1 is linearly ubiquitinated on at least two lysine residues by LUBAC. The linear ub of STAT1 inhibits its ability to signal by IFN-alpha, an antiviral cytokine. As a part of the mechanism, the linear ub of STAT1 prevents its interaction with IFNAR2 to inhibit IFN signaling. Furthermore, the authors also report that the deubiquitinase, Otulin removes the linear ub chains from STAT1 to restore the IFN signaling. Finally, the authors utilize Hoil-1+/- mice to study the biological significance of their study, and these mice exhibit reduced viral replication presumably due to the enhanced IFN-STAT1 signaling.

The study is very interesting, and using highly sophisticated biochemical approaches, the authors clearly established a novel posttranslational modification on STAT1 to regulate the IFN signaling. IFN is an antiviral cytokine; however, excessive IFN signaling can lead to undesired outcomes including autoimmune reactions and diseased states. Therefore, this study uncovers a new mechanism that may help develop new therapy based on the linearly ubiquitinated STAT1. Although the mechanistic part of the study is

quite solid, the *in vivo* studies are relatively weaker. The authors should consider the following points to improve their study:

Thank you so much for these good comments!

1. The lack of a STAT1 knock-in mouse harboring mutated lysines does not allow the authors to directly test the new mechanism on the biological functions of IFN. Therefore, the phenotype in *Hoil-1*^{+/-} mice may not be completely due to this new mechanism. The authors should find a way to establish this *in vivo* to significantly strengthen their study. Is it possible to address this partly *ex vivo* in the KO cells, isolated from *Hoil-1*^{-/-} and/or *Hoip*^{-/-} cells, by expressing STAT1 Wt or the lysine mutants. Also, the authors claim that *Hoil-1*^{-/-} are not available; however, Tokunaga et al (2009) reported these mice.

Thanks a lot for the reviewer's good comment! To further demonstrate the regulation of STAT1-lysine mutants on IFN response *in vivo*, we used a bone-marrow chimeras mouse model. This is also a suggestion from the editor. We first got the bone marrow from *Stat1*^{-/-} donor mice and infected them with the lentiviruses packaged with GFP-STAT1 (WT or K511/652R mutants). Then the bone marrow cells were injected into the BALB/c recipient mice. After ten days, these mice were administrated with mIFN β and then the *in vivo* IFN responses (including STAT1 activation and downstream ISGs expression) in mouse bone marrow were determined. These new results further suggested that STAT1 linear ubiquitination at Lys511/652 restricts IFN responses *in vivo*. Please see the new Fig. 8a, 8b, 8c.

In addition, according to the reviewer's good suggestion, we got MEF cells from the *Hoil-1*^{+/+} and *Hoil-1*^{+/-} mice, and then these cells were transfected with STAT1-WT or STAT1-K511/652R mutants (DM), followed by mIFN β treatment. The results showed that although mutation of Lys511/652 of STAT1 dramatically enhanced IFN-I-induced STAT1

activation and ISGs expression in *Hoil-1^{+/+}* MEF cells, STAT1-K511/652R mutants largely lost the ability to promote IFN-I response in *Hoil-1^{+/-}* MEF cells, suggesting that the effects of Lys511/652 of STAT1 on host IFN-I antiviral response are associated with HOIL-1L in cells. Please see the new Fig. S7c, S7d.

As to *Hoil-1^{-/-}* mice, we think that complete deletion of HOIL-1L is lethal. The mice reported by Tokunaga et al (2009, *Nat Cell Biol*) were constructed by replacing exon 7 and part of exon 8, and therefore there is a possibility that the mouse cells still express the partial fragments (exon 1-6) of HOIL-1L proteins, which could make the mice survive. Another example from Peltzer et al (2018 *Nature: LUBAC is essential for embryogenesis by preventing cell death and enabling haematopoiesis.*) demonstrated that *Hoil-1^{-/-}* mice died around embryonic day (E) 10.5, since they constructed the mice by targeting exons 1 and 2 of the *Hoil-1L* gene. Consistently, in our hands we cannot get even one *Hoil-1^{-/-}* mouse in the past two years by crossing *Hoil-1^{+/-}* mice. Thus, in this study we used the *Hoil-1^{+/-}* mice to analyze the effects of linear ubiquitination. Here we also used a bone-marrow chimeras mouse model to analyze the effect of STAT1-WT/DM on *in vivo* IFN response in our revised manuscript.

2. *The antiviral functions of the lysine mutants should be tested in the context of IFN-treated cells in Suppl Fig 3F. These results directly test the functional contribution of STAT1 linear ub, and should be moved to the main figure.*

This is a good suggestion! We re-designed this experiment and used IFNs to treat U3A (STAT1-deficient) cells that were transfected with STAT1 (WT or lysine mutants, DM), and then analyzed the antiviral functions of both STAT1-WT and STAT1-DM (Fig. S3i).

Also, we carried out the similar experiment in *Stat1^{-/-}* mouse cells (see

the following comment - comment 3 of the reviewer). The results confirmed that STAT1-Lys511/652 mutants mediate stronger IFN antiviral activity. According to the reviewer's suggestion, we have moved the data to the main figure. Please see the new Fig. 3k.

3. Also, the antiviral functions should be tested in STAT1^{-/-} mouse cells expressing the STAT1 mutants, to generalize the effects.

Please see the above experiments. We tested the antiviral functions in Stat1^{-/-} mouse cells expressing STAT1-WT or STAT1 Lys511/652 mutants, and moved the data to the main figure.

4. Virus infection induces IFN-beta in all cell types whereas IFN-alpha expression is restricted primarily to the myeloid cells. It would be relevant to also show that the new mechanism is valid in IFN-beta treated cells as well to broaden the significance of the study.

Thanks! According to the reviewer's good suggestion, we carried out a series of experiments to observe the linear ubiquitination regulation of IFN- β antiviral response. Please see the new Fig. 2g, 2h, 5b and Fig. S2b, S3g.

5. Many viruses (including SeV) inhibit phosphorylation of STAT1 and, therefore, it would be interesting to see whether an unphosphorylated STAT1 mutant can undergo the new regulation. This is to support the claim that viruses may use this mechanism to evade IFN responses, as well as the transcriptional functions of unphosphorylated STAT1.

This is a good suggestion! We actually have shown that an

unphosphorylated STAT1 mutant (STAT1-Y701F) undergoes comparable levels of linear ubiquitination modifications with STAT1-WT in cells (Fig. S4c). To better address the reviewer's comment, here we further performed several experiments. We noticed that viral infection can also upregulate linear ubiquitination of STAT1-Y701F (new Fig. S6e), which is consistent with our above analysis demonstrating that the Tyr701 residue is not required for STAT1 linear ubiquitination. However, further mutation of Lys511/652 on the base of STAT1-Y701F (STAT1-Y701F-K511/652R mutants, YF-DM) abolished virus-induced linear ubiquitination of STAT1 (new Fig. S6f). Moreover, although mutation of Lys511/652 on STAT1-WT promoted virus-mediated expression of ISGs, mutation of Lys511/652 on STAT1-Y701F cannot enhance virus-induced ISGs expression (new Fig. S6g). Consistent with these observations, mutation of Lys511/652 on STAT1-Y701F cannot enhance cellular antiviral response (new Fig. S6h). Taken together, these findings suggested that both Lys511/652 (linear ubiquitination) and Tyr701 (phosphorylation) are involved in linear ubiquitination-mediated regulation of IFN response to viral infection.

6. Please provide a summary model for the study.

Done. We have added a summary model in the new Figure 8d. Thanks!

7. Minor point: why STAT1 appears as a doublet in some of the panels?

Thanks for this comment! As shown in the old version of our manuscript (please see Line 647 and Line 654 in Methods), we have two different STAT1 antibodies in our lab, which have been proved to be specific and efficient by many studies in literature. One is from the Santa Cruz (sc-464), which is relatively cheap and good for immunoprecipitation analysis. This sc-464 antibody usually results in a strong STAT1-p91 band and a very weak STAT1-p84 band. The other antibody is from the CST (Cell Signaling Technology, #9172), which is relatively expensive and usually results in two clear STAT1 (p91 and p84) bands. Thus, the doublet STAT1 bands in some of the panels could result from the CST (#9172) antibody. Thanks!

REVIEWERS' COMMENTS:

Reviewer #1 (Remarks to the Author):

The authors have adequately addressed my concerns during the revision. I only find one small issue in the figures. In STAT1-IP group of Figure 4C, why lane 2 and lane 3 are labeled similarly? Please check it. After finishing this modifying, I think this work can be published in Nature Communication.

Reviewer #2 (Remarks to the Author):

The authors have responded to all of my previous concerns and the manuscript is not very solid. However, one remaining concern I have with the new experiment, which was also proposed by the editors, reported in Fig 8. It is not clear why the transduced bone marrow was not injected to Stat1^{-/-} mice, instead of the BALB/C mice, which also express Wt Stat1.

REVIEWERS' COMMENTS:

Reviewer #1 (Remarks to the Author):

The authors have adequately addressed my concerns during the revision. I only find one small issue in the figures. In STAT1-IP group of Figure 4C, why lane 2 and lane 3 are labeled similarly? Please check it. After finishing this modifying, I think this work can be published in Nature Communication.

Thanks for your careful checking! We have revised the label in lane 2. We greatly appreciate your nice help!

Reviewer #2 (Remarks to the Author):

The authors have responded to all of my previous concerns and the manuscript is not very solid. However, one remaining concern I have with the new experiment, which was also proposed by the editors, reported in Fig 8. It is not clear why the transduced bone marrow was not injected to Stat1^{-/-} mice, instead of the BALB/C mice, which also express Wt Stat1.

Thanks for your comment! When we performed the preliminary experiments for Bone Marrow Chimeras, we actually considered two possible recipient mice, BALB/c and 129S6 (*Stat1*^{-/-}). By discussing with Dr. Qiao Cheng (Institute of Blood and Marrow Transplantation of Soochow University), we realized that some gene knockout mice are not suitable to receive high dose of X-Ray irradiation and subsequent bone marrow transplantation procedures, because knockout of certain genes could result in intensive sensitization (very weak mouse status and even death) to irradiation. As we know, knockdown of STAT1 leads to growth suppression and strong sensitization to radiation (*Pitroda, S.P., et al, BMC Med 7, 68 (2009)*). Thus, we selected BALB/c mice to perform Bone Marrow Chimeras experiments. Given that we aim to observe the effects of STAT1 gene mutation on IFN signaling in the mice under normal conditions (WT), we believe that the WT-BALB/c mice are good for our Bone Marrow Chimeras experiments. The bone marrow cells in the WT-*Stat1* recipient mice will be replaced by the donor bone marrow with GFP-STAT1 (WT or K511/652R) during the Bone Marrow Chimeras experiments, and thereby do not hinder our study aim.